**Global warming will largely increase waste treatment CH₄ emissions in Chinese Megacities:**
**insight from the first city scale CH₄ concentration observation network in Hangzhou city,**
**China**
Cheng Hu[1,2], Junqing Zhang[1], Bing Qi[3,4*], Rongguang Du[3*], Xiaofei Xu[4], Haoyu Xiong[5], Huili
Liu[1], Xinyue Ai[1], Yiyi Peng[1], Wei Xiao[2]
[1] College of Biology and the Environment, Joint Center for sustainable Forestry in Southern China,
Nanjing Forestry University, Nanjing 210037, China
[2] Collaborative Innovation Center on Forecast and Evaluation of Meteorological Disasters
(CIC-FEMD), Nanjing University of Information Science & Technology, Nanjing, China
[3] Hangzhou meteorological bureau, Hangzhou 310051, China
[4] Zhejiang Lin'an Atmospheric Background National Observation and Research Station, Hangzhou
311300, China
[5] College of Environment, Zhejiang University of Technology, Hangzhou 311300, China

22 *Corresponding authors: Bing Qi (bill_129@sina.com), Rongguang Du (drg1998@163.com).

29 To be submitted to: *ACP*

**Abstract:**

Atmospheric $CH_4$ is the second largest anthropogenic contributor to global warming. However, its emissions, components, spatial-temporal variations and projected changes still remain large uncertain from city to national scales. $CH_4$ emissions from waste treatment (including solid waste landfills, solid waste incineration and sewage) account for >50% of total anthropogenic $CH_4$ emissions at city scale, and considering the high temperature sensitivity of $CH_4$ emission factors (EFs) for the biological processes-based sources such as waste treatment, large bias will be caused when estimating future $CH_4$ emissions under different global warming scenarios. Furthermore, the relationships between temperature and waste treatment $CH_4$ emissions have only been conducted in a few site-specific studies and lack the representativity for whole city, which contains various biophysical conditions and shows heterogeneous distribution. These above factors cause uncertainty in the evaluation of city scale $CH_4$ emissions (especially from waste treatments) and projected changes still remain unexplored. Here we conduct the first tower-based $CH_4$ observation network with three sites in Hangzhou city, which is located in developed Yangtze River Delta (YRD) area and ranks as one of the largest megacities in China. We found the *a priori* total annual anthropogenic $CH_4$ emissions and those from waste treatment were overestimated by 36.0% and 47.1% in Hangzhou city, respectively. In contrast, the total emissions in the larger region, such as Zhejiang province or the YRD area, were slightly underestimated by 7.0%. Emissions from waste treatment showed obvious seasonal patterns following air temperature. By using the linear relationship constructed between monthly waste treatment $CH_4$ emissions and air temperature, we find the waste treatment EFs increase by 38%~50% with temperature increases of $10^{\circ}C$. Together with projected temperature changes from four climate change scenarios, the global warming induced EFs in Hangzhou city will increase at the rates of 2.2%, 1.2%, 0.7% and 0.5% per decade for IPCC AR5 (International Peace Cooperation Center, the fifth assessment report) RCP (Representative Concentration Pathway)8.5, RCP6.0, RCP4.5 and RCP2.6 scenarios, respectively. And the EFs will finally increase by 17.6%, 9.6%, 5.6%, and 4.0% at the end of this century. Additionally, the derived relative changes in China also show high heterogeneity and indicate large uncertainty in projecting future national total $CH_4$ emissions. Hence, we strongly suggest the temperature-dependent EFs and the positive feedback between global warming and $CH_4$ emissions should be considered in future $CH_4$ emission projections and climate change models.

**Keyword**: $CH_4$ emissions, waste treatment, observation network, global warming

## 1. Introduction

As the second largest anthropogenic greenhouse gas, the reduction of $CH_4$ emissions is considered an effective way to mitigate future climate change on short timescales (Henne et al., 2016; Lin et al., 2021). Accurate estimation of $CH_4$ emissions from its main sources is the basis of policy making. However, recent studies find there still remain large uncertainties for its total emissions, components, spatial-temporal variations and projected changes at city scale especially for megacities in China (USPA 2013; Cai et al., 2018; Lin et al., 2021). $CH_4$ emission from waste treatment (mainly including sewage and solid waste by landfills and incineration) ranked as the world's third largest anthropogenic source after fuel exploitation and livestock, and was responsible for ~13% of global anthropogenic $CH_4$ emissions of 371 ($\pm$26) Tg $a^{-1}$ (Lu et al., 2021). It also ranked as the fourth largest anthropogenic source in China, the biggest anthropogenic $CH_4$ emitting country, and accounted for ~14% of national total anthropogenic emissions of 65 ($\pm$22) Tg $a^{-1}$ (Saunois et al., 2020; Lu et al., 2021; Chen et al., 2022). Furthermore, its contribution is even larger than 50% at city scale especially for megacities, where both active and closed household waste (including landfills and waste water systems) are located and found as super emitters (Williams et al., 2022; Maasakkers et al., 2022). A large number of Chinese landfills were constructed in suburbs more than 5-10 years ago and most landfills have no gas collection systems, with the urban area expanding in recent decades, the locations of many landfills are now within the urban scope (Zhejiang Statistical Yearbook 2018-2019). In addition, the decreasing area of the agricultural sector (rice paddies and husbandry) in megacities also makes their emissions negligible when compared with waste treatment. Therefore, accurate quantification of $CH_4$ emissions from waste treatment in urban area becomes increasingly important.

Although some progress has been made in measuring site scale $CH_4$ emissions from waste treatment, the estimated emissions still show large discrepancies due to many factors such as the amount of waste and its composition, relative proportions of landfills and incineration, degradable organic carbon ratio, $CH_4$ oxidation efficiency, and landfill gas collection, and meteorological conditions including temperature, water content, atmospheric pressure (Masuda et al., 2018; Cai et al., 2018; Zhao et al., 2019; Hua et al., 2022; Bian et al., 2022; Maasakkers et al., 2022; Kissas et

al., 2022).

Furthermore, $CH_4$ emissions from sewage and landfills result from microbial processes especially
from methanogens, and their emission factors (EFs) are highly sensitive to temperature. These
available studies were mainly conducted at some specific sites with measured EFs varying widely
(Du et al., 2017; 2018; Cai et al., 2014; 2018; Zhao et al., 2019; NBSC, 2015; Wang et al., 2015;
Florentino et al., 2010; Tolaymat et al., 2010; Hua et al., 2022). The lack and discrepancies of
detailed information for all the above factors and their uncertainties have led to considerable
difficulty in estimating $CH_4$ emissions for most-to-date inventories (Höglund-Isaksson, 2012;
USEPA et al., 2013; Cai et al., 2018; Lin et al., 2021; Maasakkers et al., 2022).

China, the developing country with the largest anthropogenic $CH_4$ emissions, is expected to
increase its emissions because of projected rapid economic development, urbanization and
generated waste (Cai et al., 2018). The increase of waste treatment emissions in East China was
also found as the second largest sector in driving national total anthropogenic $CH_4$ emissions since
2000 (Lin et al., 2021). In addition, the mitigation potential of waste treatment in developing
countries is thought to be four times that of developed countries (USEPA, 2013). Therefore,
mitigating $CH_4$ emissions from waste treatment in China is a robust and cost-effective way to
reduce total national anthropogenic greenhouse gas emissions.

Many previous studies have estimated the waste treatment $CH_4$ emissions for China by both
"bottom-up" and "top-down" approaches, with results varied by 2.5-fold from 4.3 to 10.4 Tg $CH_4$
$yr^{-1}$, and accounted for 8.1%~24.2% of national total anthropogenic $CH_4$ emissions (USEPA 2013;
Peng et al., 2016; Miller et al., 2019; Lin et al., 2021; Lu et al., 2021; Chen et al., 2022). For these
"bottom-up" approaches, the high uncertainties were directly attributed to omission of many small
point sources and discrepancies of observed site-specific EFs, which varied largely by climate and
management technology such as the efficiency of gas collection systems (Zhao et al., 2019; Hua et
al., 2022). Previous studies most commonly used the EDGAR (Emission Database for Global
Atmospheric Research) inventory, using the IPCC recommended default EF values of 15.0%

(Höglund-Isaksson, 2012; Lin et al., 2021; Bian et al., 2022), but this value is around 5-7 times of EFs used in China by Zhang and Chen et al. (2014). A recent study comparing waste treatment $CH_4$ emissions among different inventories also reported that the EDGAR v5.0 and CEDS (Community Emissions Data System) inventories were 21~153% higher than other inventories, and EDGAR v5.0 tended to assign more emissions in urban areas especially for provincial capitals. In addition, emissions from wastewater were found to be overestimated by higher emission factors or chemical oxygen demand (Peng et al., 2016; Lin et al., 2021).

And for the "top-down" atmospheric inversion approaches, a few studies constrained anthropogenic sources including waste treatment, where the most widely used concentrations were from satellite observations (Miller et al., 2019; Lu et al., 2021; Chen et al., 2022). The satellite observations have the advantage of easy data access and global coverage. But as already noted, the emissions constraint results are highly dependent on availability of observed concentrations, which are largely influenced by weather conditions and cloud coverage. As was illustrated in a recently published study by Chen et al. (2022), although the numbers of grid cell ($0.25^o \times 0.3125^o$) based year-round satellite observations were more than 1000 in north China, the available numbers were less than 10 (even including grid cells without any observations) in most of central, west, east and south of China. Such sparse distribution of available data may not provide robust constrains on waste treatment emissions for some Chinese cities without enough observations, especially considering waste treatment is co-located with high population density megacities in the developed area of east and south of China. Furthermore, there should be large temperature induced monthly variations for waste treatment $CH_4$ emissions (Börjesson et al., 1997), but almost all satellite-based inversions were conducted at annual scale without seasonal variations. Besides, given the strong influence from atmospheric pressure on landfill $CH_4$ emissions (Kissas et al., 2022), satellite observations are too sparse to be up-scaled to estimate annual total because satellite observations are mostly available only on clear-sky conditions and cannot represent atmospheric pressure and $CH_4$ emissions on cloudy or rainy days. There was only one recent study which focused on urban waste treatment $CH_4$ emissions, it found annual $CH_4$ emissions from four cities were 1.4 to 2.6 times larger than inventories in India and Pakistan, where landfills

contributed to 6~50% of total emissions and indicated large bias of our understanding of waste
treatment $CH_4$ emissions (Maasakkers et al., 2022).

The tower-based atmospheric inversion approach, which is based on hourly atmospheric
concentration observations within the planetary boundary layer, can be used independently to
constrain $CH_4$ emissions and its main components. Besides, compared with "bottom-up"
approaches, the "top-down" method can avoid using the factors that lead to large uncertainties in
$CH_4$ emissions especially from waste treatment. And to our best knowledge, there are few
tower-based observation inversion studies which focus on waste treatment emissions at city scale
or much larger regional scales especially in China. Only one study in Los Angeles, U.S.A. used
tower-based $CH_4$ concentration and found the influence of a landfill site closure on $CH_4$ emissions,
which was not included in *a priori* inventory (Yadav et al., 2019). In addition, the influences of
global warming on city scale (or higher regional scale) emissions are still unclear and have not
been considered in future emission projections (USEPA 2013; Cai et al., 2018). In general,
previous studies which predicted future waste treatment $CH_4$ emissions only used activity data
changes, without considering climate change on the EFs. Considering the potential high sensitivity
of waste treatment $CH_4$ emissions on the projected global warming, how these emissions will
change with increasing temperature is still unknown, especially within megacities where more
waste is generated and the urban heat island effect will lead to much stronger warming climate
(Zhang et al., 2022).

Here, we established three tower-based $CH_4$ concentration observation sites in Hangzhou city, one
of the largest megacities in China. To our best knowledge, it is the first city-scale tower-based $CH_4$
concentration observation network in China. We present our work on urban $CH_4$ emissions
inversion and aim to (1) constrain $CH_4$ emissions from waste treatment alongside total
anthropogenic emissions in Hangzhou city, (2) derive temperature sensitivity of waste treatment
$CH_4$ emissions at city scale and quantify the projected emission changes in future climate change
scenarios. One-year hourly $CH_4$ concentration observations from December 1st, 2020 to
November 30th, 2021 were combined with atmospheric transport model and Bayesian inversion

178 approach to constrain monthly $CH_4$ emission inventories. The constructed relationship between

179 monthly temperature and *posteriori* waste treatment $CH_4$ emissions will be used with future

180 temperature projection to quantify how the EFs will change in different global warming scenarios.


182 **2. Materials and Method**

183 **2.1 Tower-based $CH_4$ observation network and supplementary materials**

184 The city of Hangzhou, which has a population of 12.2 million and area of $1.7\times10^4$ km$^2$ (core

185 urban area of $8.3\times10^3$ km$^2$), is the capital of Zhejiang province and located in the middle of East

186 China (Figure 1a). As displayed in Figures S1-S2, the East China accounts for the majority of the

187 national total population and waste treatment $CH_4$ emissions. Hangzhou city ranked in the top 10

188 megacities in China, with annual solid waste of around 5 million tons in 2021. The tower-based

189 $CH_4$ concentration observation network includes three observation sites (Figure 1a-d), as (1)

190 Hangzhou site (120.17$^o$ E, 30.23$^o$ N, 43.2 m a.s.l.), which is located in the core urban region; (2)

191 Linan site (119.72$^o$ E, 30.30$^o$ N, 138.6 m a.s.l.), regional background site with no obvious

192 emission sources within 10 km radius; (3) Damingshan site (119.00$^o$ E, 30.03$^o$ N, 1485.0 m a.s.l.),

193 which is built on the top of a 1500 m mountain and represents background from much more

194 diluted regional emission signals. The distance is around 50 km between Hangzhou site and Linan

195 site, and around 150 km between Hangzhou site and Damingshan site. These three sites represent

196 obvious gradients from east of densely populated area (Figure 1c-d) and anthropogenic emissions

197 to west of much weaker anthropogenic influence and background conditions. Based on the wind

198 direction for the three sites, there is not any obvious difference of seasonal wind direction patterns

199 among them. The prevailing wind direction from October to February was from the north, which

200 changed to east from February to May and then changed to south during the monsoon in summer.

201

202 The air inlet heights are 25 m above ground for the Hangzhou site, 53 m at Linan and 10 m at

203 Damingshan, respectively. Atmospheric $CH_4$ concentrations at all three sites were continuously

204 measured by cavity ring-down spectroscopy analyzer (model G2301 for Hangzhou site and G2401

205 for Linan site and Damingshan site; Picarro Inc., Sunnyvale, CA). To obtain high precision

206 observations, two different standard gases were measured every 6 hours and a linear two-point fit

was used to calibrate observations, with the precision and accuracy of 2 ppb and 1 ppb. More details of the observation and calibration systems were described in Fang et al., (2014; 2022). Note that because of instrument issues at Damingshan site, there is a data gap in September-October, 2021. In general, 99.4%, 99.0%, 79.3% of hourly $CH_4$ observations were available in the whole year observation period for Hangzhou site, Linan site and Damingshan site, respectively. Meteorological observations at Hangzhou meteorological station were used to evaluate simulated meteorological fields, including air temperature at 2 m ($T_{2m}$), relative humidity (RH), downward solar radiation (S↓), wind speed (WS) at 10 m height, and planetary boundary layer height (PBLH).

Note some previous studies of city scale greenhouse gas concentration observation networks chose sites at the edge of urban borders as background in emission inversion system (i.e. Indianapolis, U.S.A., Miles et al., (2017); Los Angeles, U.S.A., Verhulst et al., (2017); Washington, DC-Baltimore, U.S.A., Lopez-Coto et al., (2020); Paris, France, Lian et al., (2021) ), but we chose to use five NOAA $CH_4$ background sites as the potential background, including UUM, TAP, YRO, YON and WLG site (Figure 1a), which were much further than the observations at Damingshan site. This strategy is based on following three reasons: (1) our footprint domain is much larger than Hangzhou city and these five sites are also located close to the edge of the model domain; (2) $CH_4$ concentrations within Hangzhou city will be influenced by seasonally varying monsoon and the monthly varying wind directions will lead to obvious changes of $CH_4$ background than only at Damingshan site; (3) our model setups can partition $CH_4$ enhancements from within Hangzhou city and other regions.

The projected climate data from four RCP (Representative Concentration Pathway) scenarios (RCP8.5, RCP6.0, RCP4.5 and RCP2.6) by MRI-CGCM3 model were downloaded from World Data Center for Climate (WDCC, https://www.wdc-climate.de/ui/), where annual air temperature at 2m was used from years 2021 to 2100. The most recent population density data for Hangzhou city is for the year 2019 and was downloaded from Chinese national resource and environmental science and data center.

**2.2 WRF-STILT model setup**

The WRF-STILT (WRF: Weather Research and Forecasting, version 4.2.2, and STILT: Stochastic Time-Inverted Lagrangian Transport) model was used to simulate hourly footprint and $CH_4$ enhancement, see more details in Hu et al. (2019; 2021). Domain setups are displayed in Figure 1a, with the outer nested domain (Domian-1, 27 km×27 km grid resolution) covering eastern and central China, and the inner domain (Domain-2, 9 km×9 km grid resolution) covering the YRD area. The physical schemes used in the WRF model are the same as in our previous studies for the YRD domain (Hu et al., 2019; 2021). The simulated $CH_4$ concentration is the sum of background and enhancement, where the enhancement is calculated by multiplying all $CH_4$ flux with hourly footprint that represents the sensitivity of the concentration changes to its regional sources/sinks with spatial resolution of $0.1^o \times 0.1^o$. To better quantify $CH_4$ components at each site, $CH_4$ enhancements from different regions and sources are also tracked and separately simulated. Besides, we should note the $CH_4$ background is important in simulating $CH_4$ concentrations and atmospheric inversion. We will choose $CH_4$ background from the five background sites based on monthly footprint as discussed in Section 3.1.

The most recent inventory of Emission Database for Global Atmospheric Research (EDGAR v6.0), which has 20 categories, and WetCHARTs ensemble mean were used as the *a priori* anthropogenic and natural $CH_4$ emissions. We should note there are many $CH_4$ inventories for some developed regions and countries (i.e. France, U.S.A., Germany) with high spatial resolutions. The reasons to choose EDGAR as *a priori* anthropogenic emissions are: (1) for all available $CH_4$ inventories that covered China, the spatial resolution of EDGAR ($0.1^o \times 0.1^o$) is the highest, and it provides the most up-to date results; (2) most previous studies that constrain emissions by atmospheric inversion studies also chose EDGAR, and our results can be directly compared with previous studies; (3) the preliminary simulation of $CH_4$ concentrations showed generally good performance with observations, indicating its spatial distributions in Hangzhou city has relatively small bias even with a potentially large bias for magnitude, which will be constrained by our atmospheric inversion method.

The main sources of $CH_4$ emissions in Hangzhou city include SWD_LDF (solid waste landfills),

WWT (waste water handling), SWD_INC (solid waste incineration), PRO (all processes related to
fuel exploitation from coal, oil, and natural gas, including extraction, transportation, refining,
distribution as list in IPCC database (https://www.ipcc-nggip.iges.or.jp/EFDB/find_ef.php), RCO
(energy for buildings, mainly containing nature gas escaping from household use) and AGS
(agricultural soils). We found emissions from SWD_LDF, WWT and SWD_INC were simply
assigned in the same locations in EDGAR inventory, and hence combined them as waste treatment.
For the $CH_4$ emissions from wetland, we used WetCHARTs ensemble mean with spatial resolution
of $0.5^o$ at monthly average (Bloom et al., 2017). Considering WetCHARTs treats rice paddies
(main source as AGS) as one wetland type, AGS in EDGAR was excluded and we assume
WetCHARTs represent all wetland $CH_4$ emissions as natural wetland and rice paddies.

2.3 **Bayesian inversion framework**
The Scale Factor Bayesian inversion (SFBI) approach was applied to interpret the atmospheric
$CH_4$ concentration (or enhancement) variations in terms of quantitative constraint on all $CH_4$
sources. The relationship between observed and simulated $CH_4$ concentrations (or enhancement)
can be expressed as follows in Equation 1:
$$y = K\Gamma + \varepsilon \quad (1)$$

Where y is the observed $CH_4$ concentration (or enhancement), K corresponds to simulated
enhancements from all categories, $\Gamma$ is the state vector to be optimized and consists of *posteriori*
SFs for corresponding categories in K, and $\varepsilon$ is the observing system error.

The optimal solution to derive *posteriori* SFs is to minimize a cost function J($\Gamma$), which represents
the mismatch between $CH_4$ observations and simulations and the mismatch between *posteriori* and
*a priori* SFs (Miller et al., 2008; Griffis et al., 2017). The cost function J($\Gamma$) can be expressed as:
$$J(\Gamma) = \frac{1}{2}\left[(y - K\Gamma)^T S_e^{-1}(y - K\Gamma) + (\Gamma - \Gamma_a)^T S_a^{-1}(\Gamma - \Gamma_a)\right] \quad (2)$$

where $S_e$ and $S_a$ are the constructed error covariance matrices for observations and the *a priori*
values, and $S_e$ consists of measurement and model errors. Here each element in *a priori* SFs $\Gamma_a$
is treated as 1. Therefore, the solution for obtaining the *posteriori* SFs is to solve $\nabla_\Gamma J(\Gamma) = 0$,
and is given by,

$$\Gamma_{\text{post}} = (K^T S_e^{-1} K + S_a^{-1})^{-1}(K^T S_e^{-1} y + S_a^{-1}\Gamma_a) \qquad (3)$$

In the Bayesian inversion framework, we first need to give an estimate of the error covariance matrices and the state vector for the *a priori* and observational data. And following our previous studies conducted in East China (Hu et al., 2019; 2022). Uncertainties of 10%, 13% and 20% were assigned to the measurement errors ($S_{obs}$), the finite number of particles (500) released in the STILT model ($S_{particles}$) and uncertainty in meteorological fields ($S_{met}$), respectively.

A previous study derived uncertainties of $CH_4$ from waste treatment and other categories, which varied between 30% and 50%, these uncertainties were calculated mainly from activity data and EFs at the country scale on annual averages (Solazzo et al. 2021). We should also note $CH_4$ emissions uncertainty will largely increase as the study region size decreases, and, as stated above, the relative difference among different inventories can reach 150%. Considering the disaggregation of spatial distributions and temporal variations, $CH_4$ emission uncertainties can be much larger at urban and monthly scales. To provide robust constraints on $CH_4$ emissions in our study, we used three cases of *a priori* uncertainty combinations for different emissions in Bayesian inversion:

(1) the first case use three elements as wetland, waste treatment and all other anthropogenic sources, considering the larger seasonality of waste treatment, the uncertainties of 300% was used for waste treatment and 200% for other categories, (2) the second case have more detailed categories as wetland, waste treatment, fuel exploitation, energy for building, and the other anthropogenic sources, where the *a priori* uncertainty of 200% was used for each category, (3) the third case has the same categories as case 1 but uses a different *a priori* uncertainty for waste treatment of 200%. The averages of all three cases are used as final *posteriori* SFs and the largest difference between each of three cases is used as the final uncertainty.

**3. Results**

**3.1 Atmospheric $CH_4$ observations**

We first display the hourly $CH_4$ concentrations from our three tower-based sites and smoothed background at five sites by CCGCRV fitting method (Thoning et al., 1989) in Figure 2a. The

hourly observations at three towers show similar temporal variations but with different amplitudes.
Observations at Hangzhou site vary between 2000 ppb and 2800 ppb, and were much larger than
both Linan site and Damingshan site. Their monthly averages are also compared in Figure 2b, and
results show the monthly $CH_4$ vary between lowest 2106.3 ppb in July and highest 2225.0 ppb in
September (annual mean of 2159.9 ppb) at Hangzhou site, lowest 2023.3 ppb in July and highest
2132.0 ppb in September (annual mean of 2086.7 ppb) at Linan site, the lowest 1955.5 ppb in July
and without observations in September at Damingshan site (annual mean of 2013.4±(3) ppb,
where the uncertainty was calculated based on the assumption that monthly $CH_4$ concentration in
September and October varies between August and November), respectively. The similar trends
among the three sites can be explained by all three sites being dominated by similar atmospheric
transport processes, such as synoptic process (i.e. monsoon) and seasonally changing wind
directions as summarized above. But their surrounding emission sources are highly different,
implying the emissions of Hangzhou site should be much larger than Linan and Damingshan sites.

Because the $CH_4$ background is important in concentration simulation and emission inversion, we
also compare $CH_4$ background between five sites, where the annual averages at TAP, YON, RYO,
WLG and UUM were 1989.8 ppb, 1850.1 ppb, 1982.7 ppb, 1973.4 ppb and 1984.2 ppb,
respectively. We found the differences were generally within 20 ppb among TAP, RYO, WLG and
UUM sites (Figure 2), but there was large difference between YON site and other four sites from
May to August, which can reach to around 100 ppb. Note YON site is located in the south of East
China Sea (Figure 1a), it can be influenced by monsoon with clean air flows from the South China
Sea, which has many fewer $CH_4$ sources compared to air flows from East Asia. The $CH_4$
background at TAP site appeared slightly higher than other four sites because TAP site is located
in the coast of South Korea and can be more easily polluted by anthropogenic emissions.
Considering the large spatial difference between the $CH_4$ background sites, monthly air flows and
source footprint will be used to identify backgrounds for our observation network, with details
discussed in Supplementary Material (Section S1, Figure S3 and Table S1).


**3.2 Concentration footprint and the *a priori* emissions**

To illustrate the potential source regions of the three sites, annual averages of simulated footprints for each site are displayed in Figure 3a-c. The results show their footprint distributions were quite similar because of close distance, but we also notice there were obvious differences in the footprint strengths (i.e. the area covered by red color) with Hangzhou site > Linan site > Damingshan site. The reason why the footprint at the Damingshan site is the lowest can be explained that the observations were collected at 1500 m height, and it was not easy to receive emissions signals within boundary layer at that height. Besides, the Hangzhou site is located in the core urban area of Hangzhou city, and it will show significant diurnal variation in PBLH, especially since it has higher nighttime PBLH caused by anthropogenic heat and high buildings than grassland/farmland, which dominate Linan site and Damingshan site. Hence more air particles can remain within PBLH and generate stronger footprint.

The *a priori* EDGAR $CH_4$ emissions for total anthropogenic categories, waste treatment and its proportions are given in Figure 3d-f. Significant gradients are observed from higher emissions in the east to lower emissions in the west, which is consistent with our three tower-based sets of observations. And the $CH_4$ emissions for waste treatment indicated similar spatial distributions with urban land use and population density (Figure 1c-d). Moreover, waste treatment seems to emit $CH_4$ as area sources instead of point sources from waste treatment super plants. Although a few previous studies found limitations of EDGAR inventory to capture $CH_4$ emission patterns in some urban areas (Pak et al., 2021), here considering the fact that locations of landfills (Figure 1b-d), which is the largest anthropogenic $CH_4$ emitter in Hangzhou city, are very close to the core urban area and in high consistency with EDGAR, hence we believe the spatial patterns of EDGAR in study region to be reliable. We should note the Chinese government constructed waste separation stations in each city with density of one station for per 150~200 households (around 450~800 people), usually these waste separation stations are full with waste because domestic garbage can be generated every day, they do not have gas collection systems and can emit large quantity of $CH_4$ emissions caused by daily biomass waste as area sources (Tian et al., 2022). Besides, there is only one landfill that has gas collection systems, the reported gas collection

efficiency was less than 80%, which also indicates large quantity of $CH_4$ emissions will be directly
emitted into the atmosphere and the emissions will be influenced by climate change. These above
analyses also imply Hangzhou site can observe higher emissions from both waste treatment and
total anthropogenic emissions, which will be discussed and quantified later.

**3.3 Simulation of $CH_4$ concentrations and its components for three sites**
Comparisons between observed and simulated daily $CH_4$ concentration averages are displayed in
Figure 4a-c and hourly concentrations in Figure S4 for three sites. First, the hourly simulations in
Figure S4 show high consistency when only comparing the temporal patterns with observations,
indicating good performance of model transport simulations as confirmed in Figure S5 for
evaluating meteorological fields. But the relative variations display obvious differences among the
three sites for daily averages in Figure 4a-c. The mean bias (MB), root mean squared error
(RMSE), and correlation coefficient (R) between daily observations and *a priori* simulations were
64.1 ppb, 129.2 ppb and 0.44, respectively, for Hangzhou site; and were -6.0 ppb, 57.1 ppb, 0.50
for Linan site, 36.2 ppb, 55.6 ppb, 0.54 for Damingshan site. As for the Hangzhou site, simulated
$CH_4$ concentrations show obvious overestimation from October to April, and the overestimation is
also found at Damingshan site. We found the simulations at the Linan site showed overall good
agreement with observation, but still with slight overestimation from January to April and
underestimation from May to September. Considering the source area contributions for the three
sites are different, these differences among the three sites indicated the bias in $CH_4$ emission
largely varied from Hangzhou city to larger regional scale.

To further quantify detailed contributions from different regions and categories to each tower site,
$CH_4$ enhancements from different categories and source areas were also simulated separately for
the three sites. As displayed in Figure 4d-e, the simulated *a priori* total enhancements at Hangzhou
site, Linan site, and Damingshan site were 244.3 ppb, 100.8, and 69.0 ppb, respectively. We also
found contributions by waste treatments dominated the total enhancements but with obvious
differences among the three sites, which varied from the highest 64.2% at Hangzhou site to the
lowest 41.4% at Damingshan site. We further calculated anthropogenic contributions from

Hangzhou city (excluding wetlands because of coarser spatial resolution for Hangzhou city) and other provinces, which were 158.4 ppb at Hangzhou site, 30.7 ppb at Linan site, and 10.1 ppb at Damingshan site, respectively. And they accounted for 69.3%, 34.0%, and 16.9% of total anthropogenic enhancements at corresponding sites. These results indicate the $CH_4$ observations at Hangzhou site, which is located at the core urban region, are more influenced by local emissions (mainly for waste treatment which will be discussed later) and contain much higher enhancements than the other two sites. The relative contributions from Hangzhou city to observations at the Hangzhou site, Linan site and Damingshan site were 158.4 ppb (69.3% to total $CH_4$ enhancement), 30.7 ppb (34.0% to total $CH_4$ enhancement), and 10.1 ppb (16.9% total $CH_4$ enhancement), respectively. The relative contributions from Zhejiang province to observations at the Hangzhou site, Linan site and Damingshan site were 181.7 ppb (79.5% to total $CH_4$ enhancement), 44.3 ppb (49.0% to total $CH_4$ enhancement), and 17.9 ppb (29.9% total $CH_4$ enhancement), respectively. These different values also imply that the observations at Linan and Damingshan sites can represent $CH_4$ emissions of much larger region as Zhejiang province or YRD area than Hangzhou city (Figure 4e), and Daminshan site.

The seasonally averaged diurnal variations for both observations and simulations are also displayed in Figure 5 for the three sites. Although many previous studies only used daytime observations and simulations to evaluate *a priori* emissions bias and constrain emissions (Sargent et al., 2018; Hu et al., 2022), these studies were based on the assumption that the diurnal scaling factors used for the *a priori* emissions are right (i.e. for anthropogenic $CO_2$), or the emissions do not have obvious diurnal variations (i.e. emissions from industries or manufacturing). As concluded above, the main $CH_4$ component in Hangzhou city was waste treatment (Figure 3f), which should be highly sensitive to temperature and indicates obvious diurnal and seasonal patterns (Mønster et al., 2019; Kumar et al., 2022). And total $CH_4$ emissions will be overestimated when using daytime emissions to represent all-day averages. Further, we found strong similarities of the diurnal variations between observations and simulations for the three sites, but there are still some discrepancies especially that the observations at Linan site were generally higher than simulations from spring to autumn for both all-day and midday averages.

Hence, our preliminary conclusions were that the *a priori* $CH_4$ emissions were generally
overestimated for Hangzhou city but underestimated in the larger region of Zhejiang or YRD area.
We also found simulations were higher than observations for all seasons at Damingshan site, and it
can be explained by the complex topography around the Damingshan site, where elevations
changed from 0 m to 1600 m within the site's grid cell of 9 km ($\sim 0.1^{o}$) as displayed in Figure 1b,
and the mountain-valley wind patterns, PBLH changes can only be resolved with much higher
spatial resolutions of < 1km. Hence the use of coarse resolutions (i.e. 9 km in this study) at the
mountainous regions introduces large bias in simulating concentration and emission inversion, as
also recently found in China for $CO_2$ as "aggregation error" (Agustí-Panareda et al., 2019; Wang et
al., 2022), so observations at Damingshan site will not be used in emissions inversions in this
study.

**3.4 Constraints on anthropogenic $CH_4$ emissions**
As displayed in Figures 3f, 5a and concluded in Section 3.3, simulations using *a priori* $CH_4$
emissions show obvious overestimation especially from October to April at Hangzhou site, and
emissions were also overestimated in winter and underestimated from spring to autumn at Linan
site. Note this bias can be attributed to *a priori* emissions or meteorological simulations. Our
previous studies in YRD have evaluated the meteorological simulations by using the same
physical parameterization schemes, which showed high consistency with observations (Hu et al.,
2019; 2021; 2022; Huang et al., 2021). We also evaluated the meteorological simulations with
observations and confirmed with good model performance (Figure S5). Note PBLH simulations
are important in evaluating model performance, we only have four months of PBLH observations
(one month in each season), these hourly PBLH observations were used to evaluate the general
performance of WRF model. As displayed in Figure S6, it shows overall good performance for
both daytime and nighttime PBLH variations. Furthermore, we found there no monthly variations
in EDGAR v6.0 $CH_4$ emissions for waste treatment, which contributed 64.2% to annual $CH_4$
enhancement average and much higher in winter (Figure S7-S8). The $CH_4$ emissions from waste
treatment are produced by the microbial process, which should be affected by meteorological
conditions especially by seasonal temperature changes. Hence our assumption is that the bias in
both its seasonality and annual average lead to large overestimation/underestimation in the
simulated $CH_4$ concentration. Besides, bias in other anthropogenic emissions and wetlands can
also partly contribute to the bias of the simulated $CH_4$ concentration.

To quantify the bias sources and constrain corresponding *a priori* emissions for Hangzhou city, we
applied the scaling factor Bayesian inversion approach with three different cases as introduced in
the Method section. Instead of only using daytime $CH_4$ observations to constrain *a priori*
emissions, we choose to use all-day hourly data at Hangzhou site to constrain emissions for
Hangzhou city, for the following three reasons: (1) the enhancements contributed by Hangzhou
city at the Hangzhou site was 69.3%, and much larger than 34.0%, and 16.9% for Linan site and
Damingshan site, respectively; (2) the waste treatment dominated anthropogenic $CH_4$ emissions in
Hangzhou city, which is caused by biological process and should be temperature dependent. Since
the observed temperature varied diurnally by 20 $^{o}C$, the use of only daytime observations without
considering diurnal $CH_4$ emissions will bring significant bias when using derived daytime
emissions to represent all-day averages. The annual averages of daytime and all-day average
concentrations were 2112.4 and 2156.0 ppb at Hangzhou site, respectively, the reason why higher
emissions in daytime correspond to lower concentration than in all-day and nighttime is that lower
PBLH in nighttime will leads to higher concentration, and more comparisons between daytime
and all-day average concentrations are displayed in Figure 5 for three sites; (3) previous studies
using daytime observations were mainly conducted for regions dominated by industry or energy
production, which have much smaller diurnal variations than waste treatment as stated above
(Mønster et al., 2019; Kumar et al., 2022).

The derived monthly *posteriori* SFs for each emission source are displayed in Table 1 for
Hangzhou city. The results show that the *posteriori* SFs for waste treatment are much smaller in
winter and higher in summer, indicating obvious seasonality and the overestimation in winter was
mainly contributed by waste treatment. The annual mean *posteriori* SFs for waste treatment vary
between 0.50 and 0.56 in all three cases, illustrating overestimation at annual average for the *a*
*priori* waste treatment emissions. Besides, the annual mean *posteriori* SFs vary between 0.87 and

0.94 for the rest of the total anthropogenic categories (excluding agricultural soil), and are 0.97 for

PRO (fuel exploitation) and 0.91 for RCO (energy for building), respectively; the annual mean

*posteriori* SF is 1.05 for wetland (including agricultural soil and natural wetland). These *posteriori*

SFs for the rest anthropogenic categories and wetland indicate much smaller bias than waste

treatment. The monthly *posteriori* SFs for PRO and RCO also illustrate obvious seasonal

variations, but are still smaller than the *a priori* seasonality in the inventory (Figure S9). Although

the evaluations of hourly PBLH simulations have illustrated good performance in both daytime

and nighttime (Figure S6), we also conducted inversions by only using daytime observations to

constrain $CH_4$ emissions. Considering results from Case 2 varied between Case 1 and Case 3, here

we only display the results from Case 1 and Case 3 (Table S2), it shows similar seasonal variations

as using all all-day observations. We notice the values are larger than later, which is reasonable

because $CH_4$ emissions in daytime should be larger than all-day and nighttime emissions. In

general, *posteriori* SFs by using all-day concentration observations will be used to represent total

$CH_4$ emissions from monthly to annual scales.

To evaluate whether the *posteriori* SFs have significantly improved $CH_4$ emissions, we used these

SFs to derive the *posteriori* emissions and re-simulated hourly concentrations in Figure 6 (and

daily averages in Figure S9). Results show the hourly overestimation by using *a priori* emissions

is largely reduced by using *posteriori* emissions when compared with observations in Figure 6a-b,

and the regression slopes between daily averaged observations and simulations decrease from

1.51(±0.15) for *a priori* simulations to 0.85(±0.07) for *posteriori* simulations in Figure 6c. The

mean bias (MB), root mean squared errors (RMSE), correlation coefficient (R) between daily

observations and *a priori* simulations are 64.1 ppb, 129.2 ppb and 0.44, respectively, and these

statistics change to -22.2 ppb, 72.3 ppb and 0.58 for *posteriori* simulations. These results indicate

the *posteriori* SFs obviously decrease the bias in *a priori* emissions and are closer to observations,

when considering there are no system biases in simulated monthly PBLH.

The comparisons of monthly $CH_4$ emissions between *a priori* and *posteriori* waste treatment and

other anthropogenic sources (excluding agricultural soil) in Hangzhou city are displayed in

Figures 7a and S7. For the *a priori* inventory, there is not seasonal variations for waste treatment with constant monthly emissions of $8.67 \times 10^3$ t, and other anthropogenic sources show seasonality with much higher in winter (i.e. $5.22 \times 10^3$ t in January) than in summer (i.e. $3.06 \times 10^3$ t in August). The seasonality in *a priori* EDGAR inventory is mainly dominated by RCO (Energy for buildings), with proportions to total anthropogenic emissions changing from the highest 22% in winter to lowest 8% in summer. Such information indicates the *a priori* inventory assigned more leaks from natural gas distribution infrastructure in winter than in summer. As discussed above, constant emissions from waste treatment should be wrong because of its large temperature sensitivity, and the observed monthly temperature difference between summer and winter was larger than 25°C in Hangzhou city in study period. After including the constraints from the observed concentrations, the *posteriori* emissions for waste treatment show obvious seasonality with highest emission in July ($7.66 \pm 0.09 \times 10^3$ t) and lowest emission in February ($2.20 \pm 0.87 \times 10^3$ t). And emissions from other anthropogenic categories show much smaller seasonality (highest emission in January of $4.18 \pm 0.69 \times 10^3$ t and lowest emission in August of $2.88 \pm 0.15 \times 10^3$ t) than *a priori* emissions. In general, the annual emissions from waste treatment were $10.4 \times 10^4$ t in the *a priori* EDGAR inventory and decreased to $5.5 (\pm 0.6) \times 10^4$ t for the *posteriori* emissions, a decrease of 47.1%. The *a priori* emissions from other anthropogenic sources were $4.5 \times 10^4$ t and only slightly decreases to $4.1 (\pm 0.3) \times 10^4$ t for the *posteriori* emissions, an 8.9% decrease. The proportion of waste treatment to total anthropogenic emissions decreases from *a priori* 69.3% to *posteriori* 57.3%. To summarize, the annual total anthropogenic $CH_4$ emission (excluding agricultural soil) decreases from $15.0 \times 10^4$ t to $9.6 (\pm 0.9) \times 10^4$ t, indicating overestimation of 36.0% in Hangzhou city for the *a priori* emissions.

However, as concluded above the observations and simulations at Linan site, which represents the much larger region of Zhejiang province or YRD area, data from that site indicated slightly different results that $CH_4$ simulations were underestimated from spring to autumn and overestimated in winter (Figure 4b and Figure 5e-h). Here we used the multiplicative scaling factor (MSF) method and observations at Linan site to derive SFs at seasonal scale (Sargent et al., 2018; He et al., 2020), where we used 10 ppb as the potential $CH_4$ background uncertainty in

winter, spring and autumn, and 20 ppb in summer, see details in the Supplementary Material
(Section S2). The derived *posteriori* SFs were 0.87 (±0.08), 1.07 (±0.11), 1.19 (±0.24), and 1.16
(±0.11) for winter, spring, summer, and autumn, respectively. The results for the Linan site
showed similar seasonal variations as found for Hangzhou city and was 1.07 (±0.14) of *a priori*
anthropogenic emissions for the annual average. Our observations at Hangzhou site and Linan site
together indicate the *a priori* emissions were largely biased on both seasonal and annual scales,
and the annual anthropogenic $CH_4$ emission was largely overestimated by 36.0% in Hangzhou city,
but was underestimated by 7.0% in the larger region of Zhejiang province or YRD area.

**3.5 Temperature sensitivity of waste treatment $CH_4$ EFs and projected changes**
Although the derived *posteriori* monthly SFs on waste treatment reflected changes on emissions,
considering the monthly activity data does not have obvious monthly changes, these SFs can
mainly reflect relative variations of monthly EFs and contain meteorological dominated changes
especially for temperature. To evaluate the temperature sensitivity of its EFs, we first calculated
the normalized monthly SFs by dividing monthly SFs by annual averages (Tables 1 and S3), and
quantified the relationship between observed $T_{2m}$ and normalized SFs. Note decomposition of
organic waste by methanogens mostly takes at depth within the landfills and temperature can be
higher than at the surface, hence the temperature within landfills should be much more related to
methanogens activities and $CH_4$ emissions than $T_{2m}$. However, considering (1) we do not have
direct temperature observations under landfills, (2) $T_{2m}$ can be used as indicator of methanogens
activities, and (3) $T_{2m}$ is commonly used meteorological data that can be provided for future RCP
scenarios, hence the relationship between waste $CH_4$ emissions and $T_{2m}$ is constructed and used to
predict how will $CH_4$ EFs change in different climate scenarios. The normalized SFs illustrate
significant linear relationship with monthly $T_{2m}$ (Figure 7b), where the slopes imply that
normalized SFs (and EFs) will increase by 38%~50% with temperature increase by 10$^{o}$C at city
scale. We also analyzed the temperature sensitivity by only using daytime $CH_4$ observations and
simulations in Figure S10, it still shows strong linear relationship between normalized SFs and
$T_{2m}$, with the slopes of 0.046 and 0.060. These results are in high consistency with using all-day
observations of 0.038 and 0.050, indicating similar results of using 24 hours observations and only
using daytime observations, and less influence of simulated nighttime PBLH bias on
corresponding temperature sensitivity.

We should note the precipitation, soil water content and atmospheric pressure can also have
obvious influence on $CH_4$ emissions, and considering the fact that we have not conducted field
measurement in landfills and landfills are usually covered by metal or plastic in China to avoid the
spread of odors, hence reanalysis data cannot represent real soil water contents in these site scale
landfills. Precipitation and atmospheric pressure show obvious linear relationship with
temperature as displayed in Figure S11. They display positive linear relationship between
precipitation (affect water content) and $T_{2m}$, and negative linear relationship between monthly
averaged atmospheric pressure and $T_{2m}$. We also found negative relationship between atmospheric
pressure and normalized SFs, and positive relationship between $T_{2m}$ and normalized SFs (Figures
7b and S11). Considering air temperature always displays negative relationship with atmospheric
pressure as warmer air temperature coincides with lighter air mass and lower atmospheric pressure
in summer as displayed in Figure 11b, and colder air temperature coincides with heavier air mass
and higher atmospheric pressure in winter. Hence, the temperature can be used to represent
co-influence of both temperature and atmospheric pressure, and we only focus on the influence of
temperature on $CH_4$ emissions and will add more supporting data in following studies.

Our findings for the high sensitivity of waste treatment $CH_4$ emissions to temperature also suggest
a dramatic increase with the projection of future global warming trends. We further derived the
$T_{2m}$ trends for four different RCP scenarios as RCP8.0, RCP6.0, RCP4.5 and RCP2.6 (Figure 8a).
The results show $T_{2m}$ will increase by $0.50^{o}C$, $0.28^{o}C$, $0.16^{o}C$, $0.10^{o}C$ per decade for Hangzhou
city, respectively. These different warming trends also indicate distinct temperature-dominated
influence on future $CH_4$ EFs and emissions from waste treatment. We then used the slopes from
Figure 7b and annual temperature from 2021 to 2100 to derive relative changes of EFs in future 80
years, where observations for year 2021 were treated as the baseline year. As displayed in Figure
8b, the EFs in RCP8.5, RCP6.0, RCP4.5 and RCP2.6 scenarios will increase with the rates of
2.2%, 1.2%, 0.7% and 0.5% per decade, respectively. And $CH_4$ EFs for waste treatment will be
higher by 17.6%, 9.6%, 5.6%, and 4.0% at the end of this century.

The spatial distribution of $T_{2m}$ trends for all of China is also displayed in Figure S12, which shows
heterogeneous distribution across China for four global warming scenarios. Because East China
has high population density, with the majority of the national population (Figure S1), and is
responsible for the largest domestic garbage induced $CH_4$ emissions (Figure S2), these combined
factors indicate considerable $CH_4$ emissions changes from waste treatment in such a
temperature-sensitivity area. Considering that the temperature sensitivity of waste treatment $CH_4$
EFs is caused by microbial process at regional scales, the sensitivity can represent general
conditions of different cities or landfills. And if we assume the derived temperature sensitivity
(increase by 44% with temperature increases of $10^{o}C$ on average) is applicable for China as a
whole, especially for East China, the relative changes of waste treatment $CH_4$ EFs can be
calculated by multiplying this value by air temperature trends. The spatial distribution of global
warming induced EF changes at the end of this century is displayed Figure 9. For RCP2.6 scenario,
EFs for waste treatment will slightly increase by 4.0-6.5% in the north eastern China and increase
by 3.0-4.0% in south eastern China. The RCP6.0 also displayed heterogeneous changes in East
China, with EFs in the north eastern China increasing by 10.5-13.0% and in south eastern China
increasing by 9.0-10.5%. Relative changes in RCP4.5 and RCP8.5 are more homogeneous for East
China, which indicates EFs will significantly increase by 5.0-7.5% and 17.5-19.5%, respectively.
The largest changes will occur in West China for RCP8.5, with EFs increasing by >20.0%, but this
area has low population density and $CH_4$ emissions, and therefore these effects of global warming
can be ignored (Figure S12). Finally, we should note these derived relative changes are only
caused by global warming, and the influence of activity data, management technology and other
factors is not considered and out of the scope of this study.

**4 Discussions and implications**
Many previous studies have compared total $CH_4$ emissions and its components for different
inventories and bottom-up methods, which illustrated large uncertainty and bias at city scale and
these biases were much larger for waste treatment (Peng at al., 2016; Saunois et al., 2020; Lin et

al., 2021; Bian et al., 2022). A recent bottom-up research compared wastewater $CH_4$ EFs in China, which largely varied by four-fold in different provinces and the uncertainty in the same province were even two-fold larger than its average, implying considerable uncertainty in recent understanding of waste treatment EFs at regional scale (Hua et al., 2022). And for the national total emissions, waste treatment $CH_4$ emissions varied between 5 and 15 Tg $a^{-1}$ (Peng et al., 2016; EDGAR v6). There are also other atmospheric inversion studies in estimating China's $CH_4$ emissions (Hopkins et al., 2016; Hu et al., 2019; Huang et al., 2021; Miller et al., 2019; Lu el., 2021; Chen et al., 2022). These studies found large variations of national emissions for almost all inventories, which were mainly caused by fossil fuel exploitation, agricultural sector (livestock and rice paddies) and waste treatment. For the comparisons of waste treatment emissions, these satellite-based inversions also largely varied between 6 and 9 Tg $a^{-1}$ by 1.5-fold (Miller et al., 2019; Lu et al., 2021; Chen et al., 2022; Zhang et al., 2022).

The reported discrepancies between "bottom-up" and "top-down" approaches indicate large uncertainty in understanding China's national $CH_4$ emissions from waste treatment. And it is well known the uncertainties will increase from national scale to regional and city scales, also implying considerable uncertainties in city-scale emissions for inventories. But the atmospheric inversion approach for city scale waste treatment, which can act as an independent evaluation, is still rare not only for China but also globally. To our best knowledge, there is only one recent atmospheric inversion research focused on $CH_4$ emissions from city-scale waste treatment, which used satellite-based observation to constrain emissions from four cities in India and Pakistan, that concluded underestimation of landfills $CH_4$ emissions by 1.4 to 2.6 times for EDGAR inventory (Maasakkers et al., 2022). In our study, we found annual waste $CH_4$ emissions were overestimated by 47.1% for Hangzhou city, our findings are different from results in India and Pakistan. These differences indicate bias of waste treatment $CH_4$ emissions considerably varied in different countries and climate divisions. Our results highlight there is a large knowledge gap in understanding waste treatment emissions mechanisms and estimating urban waste treatment $CH_4$ emissions especially in China.

Different from fossil-type sources that have much smaller monthly variations, $CH_4$ emission from
waste treatment is biological processes-based source and its EFs are highly sensitive to
meteorological conditions especially for temperature. These factors lead to obvious bias in waste
treatment $CH_4$ emissions not only for annual average but also for its seasonality. Besides, although
there were a few studies that aimed to predict future $CH_4$ emissions from waste treatment, these
studies were mainly based on activity data changes without considering the EFs variations caused
by future global warming trends or only based on site-specific observations (USEPA 2013; Cai et
al., 2018; Spokas et al., 2021). Of these three cited studies, USEPA (2013) and Cai et al. (2018)
only predicted emissions changes due to changes in activity data and management technology.
And the $CH_4$ emissions for year 2030 by Cai et al. (2018) was 23.5% lower than the USEPA
(2013) estimation, which was caused by the consideration of new policies and low-carbon policy
scenarios. Spokas et al. (2021) modeled the $CH_4$ emission changes with increasing air
temperature, where $CH_4$ emissions did not show obvious changes even with temperature
increasing by ~5$^{o}$C by the end of year 2100. To our best knowledge, there are no inventories that
considered the temperature-induced changes on both seasonal variations and annual trends of $CH_4$
emissions. Hence, it is still unclear in all inventories how EFs will change with different global
warming scenarios at city scale.

A few observation-based measurements were conducted for waste treatment but only at some
specific sites with large discrepancies of EFs (Du et al., 2017; 2018; Cai et al., 2018; Zhao et al.,
2019; NBSC, 2015; Wang et al., 2015; Florentino et al., 2010; Tolaymat et al., 2010; Cai et al.,
2014; 2018). And only one of our previous studies used year-round atmospheric $CH_4$ observations
to constrain regional scale $CH_4$ emissions at Nanjing city in YRD area (Huang et al., 2021), where
it found much higher emissions of the landfilling waste in summer than in winter: $CH_4$ emissions
in July were around four times those in February. But there is no study that has quantified the
temperature sensitivity of waste $CH_4$ emissions at city scale or much larger regional scales. These
two studies in different cities confirmed temperature as the dominant factor that drives seasonal
variations of waste treatment $CH_4$ emissions. Hence our study appears as the first one that
estimated city scale waste treatment $CH_4$ emissions, its temperature sensitivity and projected
changes in different global warming scenarios. Our findings for the large sensitivity to
temperature indicate the monthly scaling factors should be considered to better represent $CH_4$
emissions and simulate atmospheric $CH_4$ concentrations.

We also note that the predictions of future climate changes are mainly based on different emitting
intensity of greenhouse gas, and $CH_4$ contributed around 20% of direct anthropogenic radiative
forcing (Seto et al., 2014). The $CH_4$ emissions in different global warming scenarios were mainly
calculated by predicting energy use data without considering the changes of EFs. In this study, we
found there should be large positive feedback between global warming and $CH_4$ emissions,
especially in the RCP 8.0 scenario where global warming induced $CH_4$ emissions from waste
treatment will increase by 17.6%. Hence the projected emissions from waste treatments and other
biological process based sources, together with positive feedback between temperature and their
emissions are strongly suggested in future climate change models. Besides, it is well known that
$CH_4$ concentration simulations are essential for modeling air pollutions (e.g. $O_3$, $NO_x$, and CO)
especially in the stratosphere (Isaksen et al., 2011; Kaiho et al., 2013). Considering that waste
treatment $CH_4$ emissions accounted for ~25% of total anthropogenic emissions (EDGAR v6.0) in
East China where severe air pollution frequently occurred, we also believe the coupling of
temperature-dependent $CH_4$ emissions and the monthly scaling factors on $CH_4$ emissions can
improve air pollution modeling in East China.

We should note that new technology and other meteorological variables can also influence waste
treatment $CH_4$ emissions. The main reason to only use temperature in this study is that we only
constrained the emissions at monthly scale in one year, and derived twelve datasets of *posteriori*
$CH_4$ emissions. Besides, temperature is considered to be the main factor in controlling monthly
and annual variations of waste treatment $CH_4$ emissions, and can be used to represent the
co-influence of other meteorological parameters such as atmospheric pressure. We will use
multiple years' $CH_4$ concentration to quantify the influence of new technology and other
meteorological variables on waste treatment $CH_4$ emissions in our following study, and we suggest
that other tracers (e.g. ethane, $^{14}CH_4$) are also important to separate $CH_4$ emissions from biological
and fossil $CH_4$ emissions.

**5 Summary and Conclusions**

To better evaluate bias for city scale anthropogenic $CH_4$ emissions and understand the sensitivity
of temperature on waste treatment $CH_4$ emissions, we used a three tower-based atmospheric $CH_4$
observation network in Hangzhou city, which is located in the developed YRD region and one of
the top 10 megacities in China. One-year hourly atmospheric $CH_4$ observations were presented
from December 2020 to November 2021. We then applied a scaling factor Bayesian inversion
method to constrain monthly anthropogenic $CH_4$ emissions and its components (especially for
waste treatments) in Hangzhou city, and also used multiplicative scaling factor method for broader
Zhejiang province and YRD area at seasonal scale.

To the best of our knowledge, our study is the first tower-based $CH_4$ observation network in China.
We found obvious seasonal bias of simulated $CH_4$ concentrations at the core urban area of
Hangzhou city, which was mainly caused by bias of waste treatment at both annual and monthly
scales. The derived *posteriori* $CH_4$ emissions display obvious seasonal variations with peak in
summer and trough in winter, which was mainly contributed by waste treatment; the *a priori*
annual waste treatment $CH_4$ emission in Hangzhou city was $10.4 \times 10^4$ t and decreased to 5.5
$(\pm 0.6) \times 10^4$ t for the *posteriori* emissions, a decrease of 47.1%. Besides, the total anthropogenic
$CH_4$ emissions (excluding agricultural soil) decreased from $15.0 \times 10^4$ t to $9.6(\pm 0.9) \times 10^4$ t,
indicating overestimation of 36.0% for the whole year of 2021. Observations at Linan site imply
that the annual $CH_4$ emissions was slightly underestimated by 7.0% for the larger region of
Zhejiang province or YRD area, which was different from the case of Hangzhou city. Additionally,
the *posteriori* monthly $CH_4$ emissions from waste treatment illustrate significant linear
relationship with air temperature, with regression slopes indicating an increase of 38%~50% when
temperature increases by $10^{\circ}C$. Finally, we found the waste treatment $CH_4$ EFs for Hangzhou city
will increase by 17.6%, 9.6%, 5.6%, and 4.0% by the end of this century for RCP8.0, RCP6.0,
RCP4.5 and RCP2.6 scenarios, respectively. The derived relative changes for whole China also
showed high heterogeneity and indicate large uncertainty in projecting future national total $CH_4$

emissions. This study is also the first one that mainly focuses on city scale temperature sensitivity of waste treatment $CH_4$ emissions from the perspective of atmospheric inversion approach. And based on above results, we strongly suggest the temperature-dependent EFs should be coupled in both recent $CH_4$ inventories and future $CH_4$ emission projections.

**Data availability:** The atmospheric $CH_4$ observations data can be requested from Cheng Hu and Bing Qi. STILT model is downloaded from http://www.stilt-model.org/, the EDGAR inventory is from https://edgar.jrc.ec.europa.eu/, and the projected climate data were downloaded from World Data Center for Climate (WDCC, https://www.wdc-climate.de/ui/).

**Acknowledgement:** Cheng Hu is supported by the National Natural Science foundation of China (grant no. 42105117) and Natural Science Foundation of Jiangsu Province (grant no. BK20200802). Wei Xiao is supported by the National Key R&D Program of China (grants 2020YFA0607501 & 2019YFA0607202). This work is also supported by Zhejiang Provincial Basic Public Welfare Research Project (LGF22D050004). We sincerely thank the detailed comments from two anonymous reviewers. We also want to express our thanks to Prof. Timothy J. Griffis from University of Minnesota, who provided many important suggestions and support for this study.

**Author contribution:** Cheng Hu and Bing Qi designed the study. Cheng Hu performed the model simulation, data analysis and wrote and revised the paper; Bing Qi and Rongguang Du conducted $CH_4$ concentration observation and meteorological data collection, and all co-authors contributed to the data/figures preparation and analysis.

**Declaration of competing interests:** The authors declare that they have no conflict of interest.

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

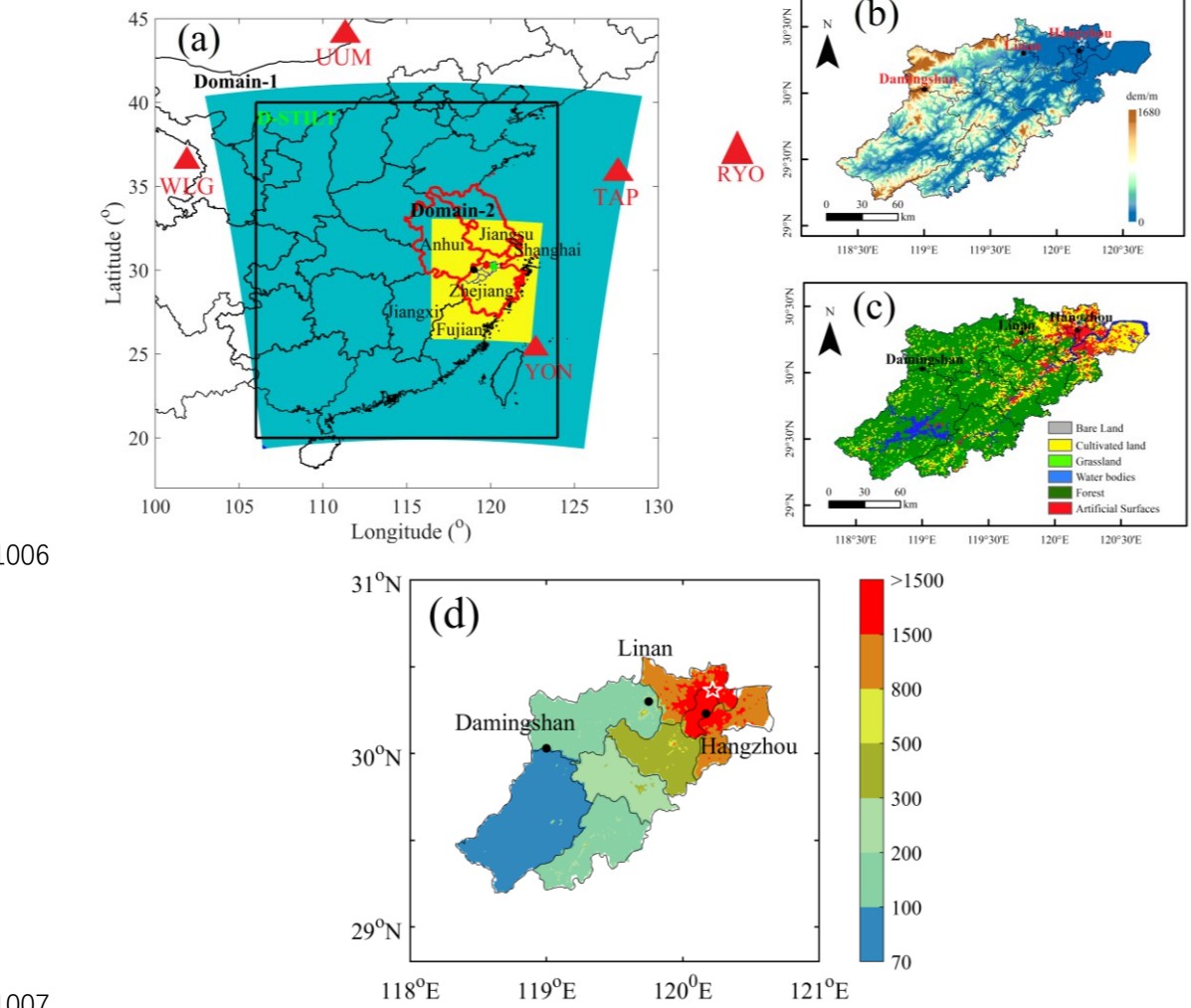


Figure 1. (a) WRF-STILT model domain setups, three CH$_4$ concentration observation sites in
Hangzhou city, and five CH$_4$ background sites, note the green, red and black dots represent
locations for Hangzhou site, Linan site and Damingshan site, respectively, Yangtze River Delta
regions is displayed in red boundary, back rectangle represents domain in STILT model, (b)
geophysical height within Hangzhou city, (c) land surface categories in Hangzhou city, and (d)
population density in Hangzhou city for year 2019, units: person per km$^2$, the location of landfills
in Hangzhou city is displayed with white star.








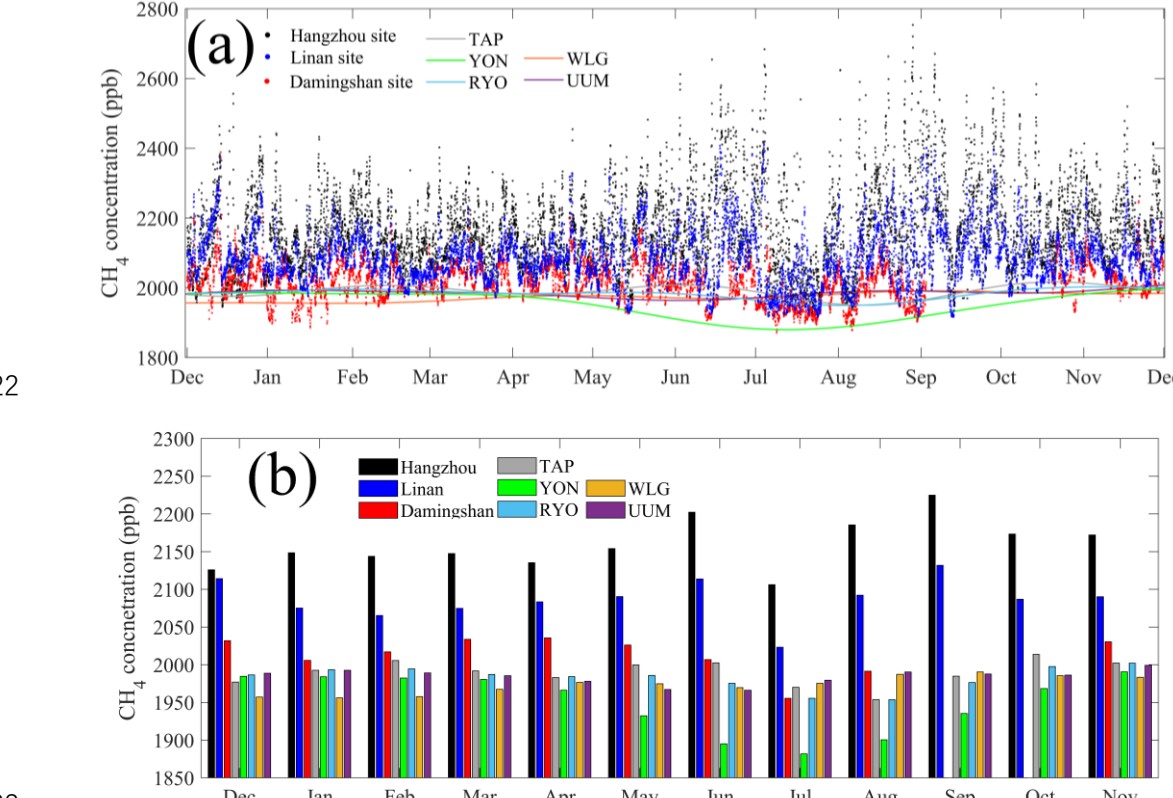


Figure 2. (a) Hourly CH$_4$ concentrations at three sites within Hangzhou city as Hangzhou site,
Linan site, and Damingshan site, and fitting CH$_4$ background based on CCGCRV regression
method at five background sites as TAP, YON, RYO, WLG and UUM, (b) monthly mean of CH$_4$
concentrations for above eight sites. Note the CH$_4$ background is smoothed by using CCGCRV
fitting method on weekly or hourly observations, which can filter large fluctuations caused by
sudden and unidentified sources


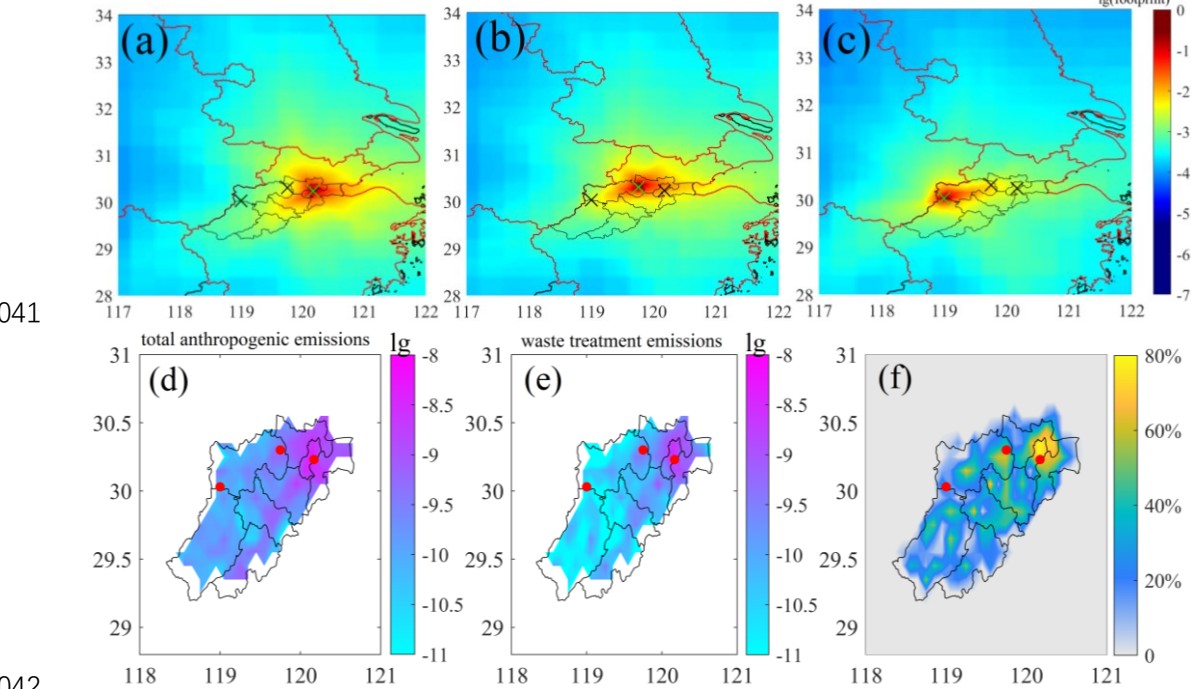


Figure 3. Annual averages of simulated footprint for (a) Hangzhou site, (b) Linan site, and (c)
Damingshan site, where the green symbol "×" indicates receptor location in each pannel, (d) total
anthropogenic $CH_4$ emissions in EDGAR v6.0 inventory, (e) waste treatment $CH_4$ emissions in
EDGAR v6.0 inventory, and (f) proportions of waste treatment to total anthropogenic $CH_4$
emissions, red dot represents three sites, units for footprint: ppm $m^2$ s $mol^{-1}$, units for emissions:
kg $m^{-2}$ $s^{-1}$. The divisions in Hangzhou city are different districts.

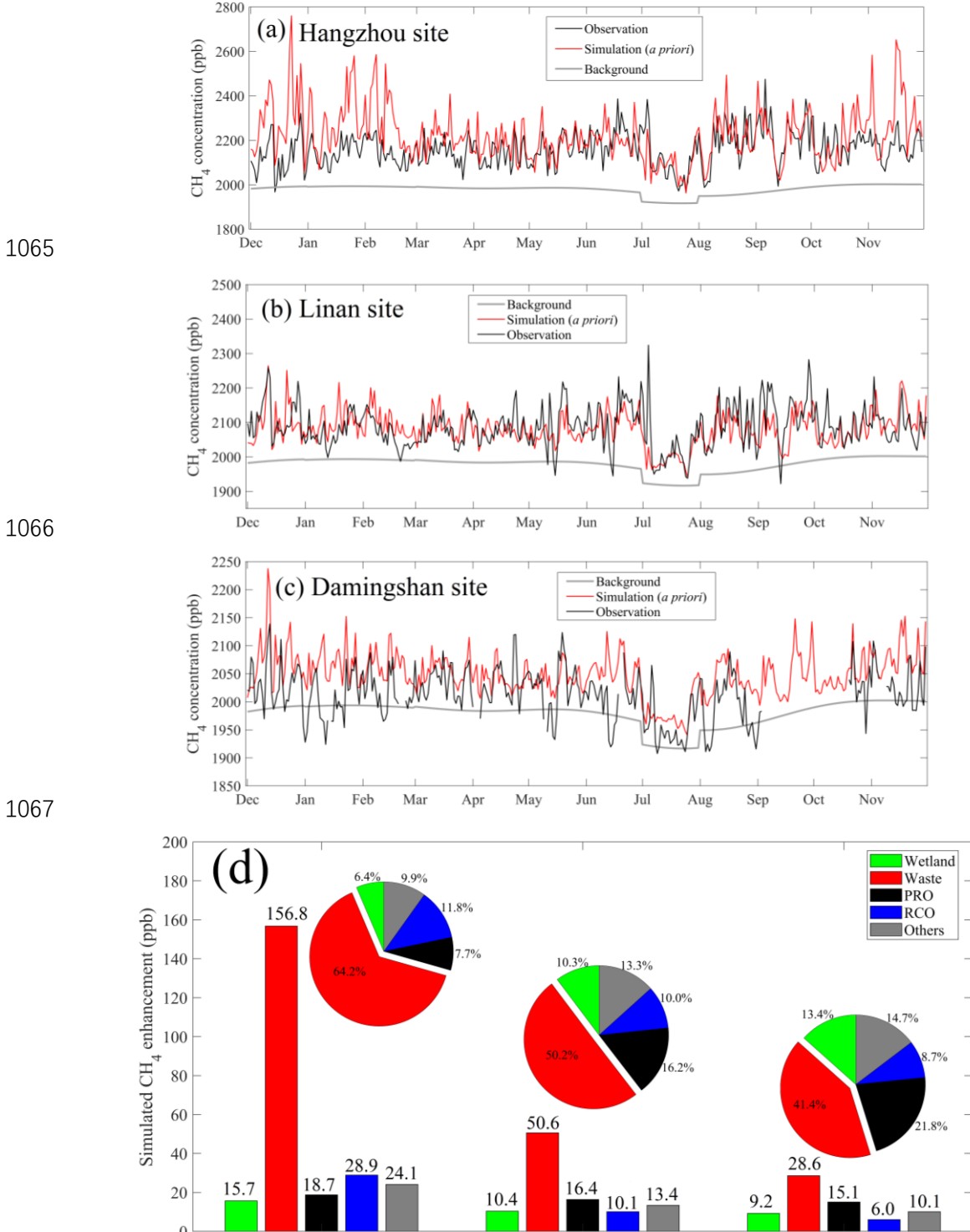





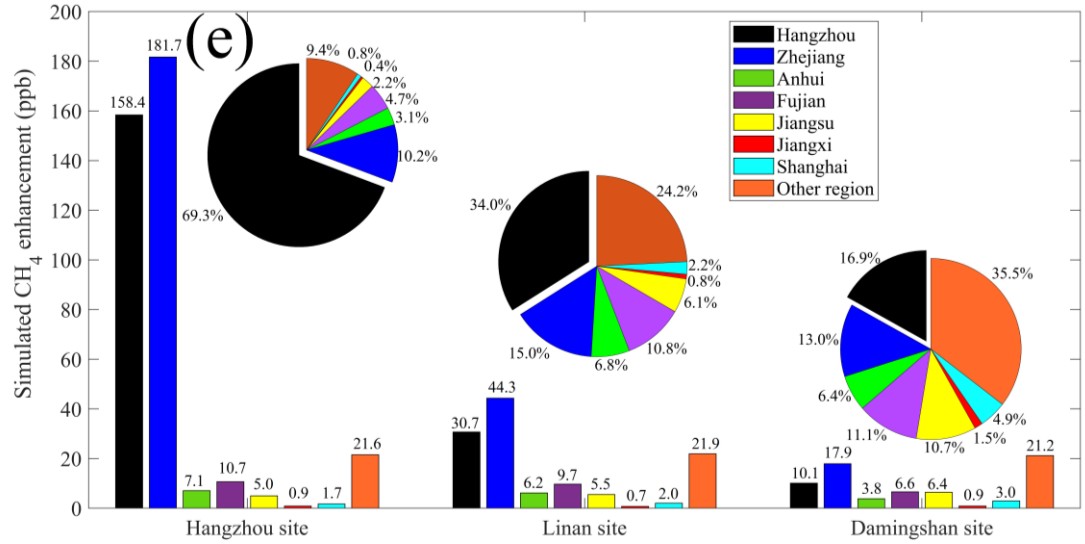

Figure 4. Comparisons between daily CH$_4$ observations and simulations for (a) Hangzhou site, (b) Linan site, (c) Damingshan site, (d) simulated CH$_4$ enhancements from main emission categories (e) simulated anthropogenic CH$_4$ enhancement from different regions and its proportions. Note the blue color for the bar charts include all contributions from "Zhejiang", including "Hangzhou"; and the blue regions in the pie charts represent rest regions of "Zhejiang minus Hangzhou".

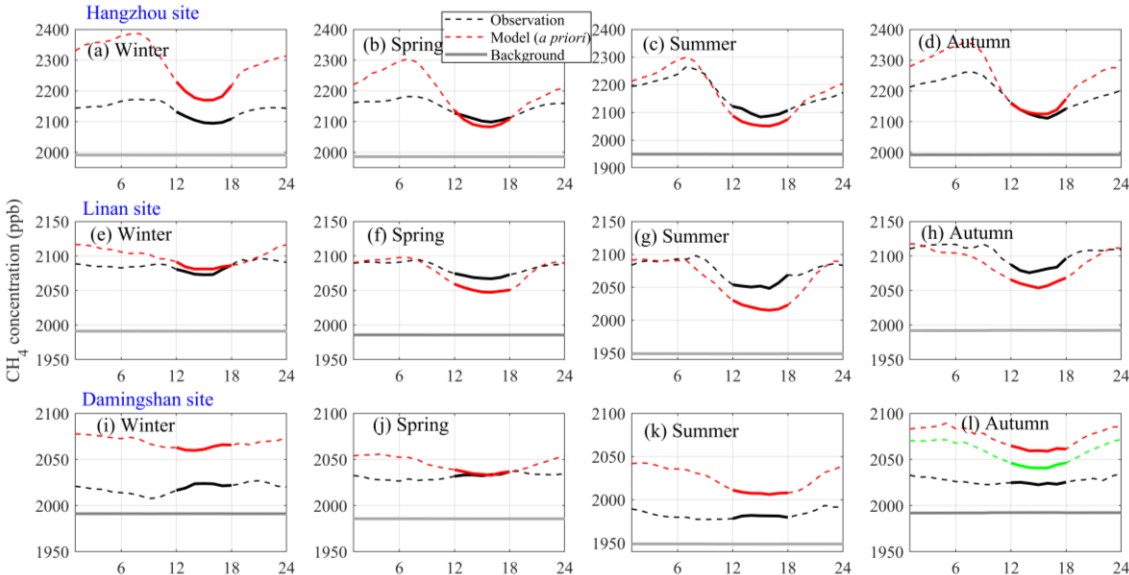

1097

Figure 5. Seasonal averaged diurnal variations for Hangzhou site in (a) winter, (b) spring, (c) summer, (d) autumn, and Linan site in (e) winter, (f) spring, (g) summer, (h) autumn, and Damingshan site in (i) winter, (j) spring, (k) summer, (l) autumn; Note because of two months of data gap in Autumn for Damingshan site, the green line is for all September-November simulations, red line only represent simulation of corresponding period for available observation data, and bold lines represents data between 12:00 and 18:00.



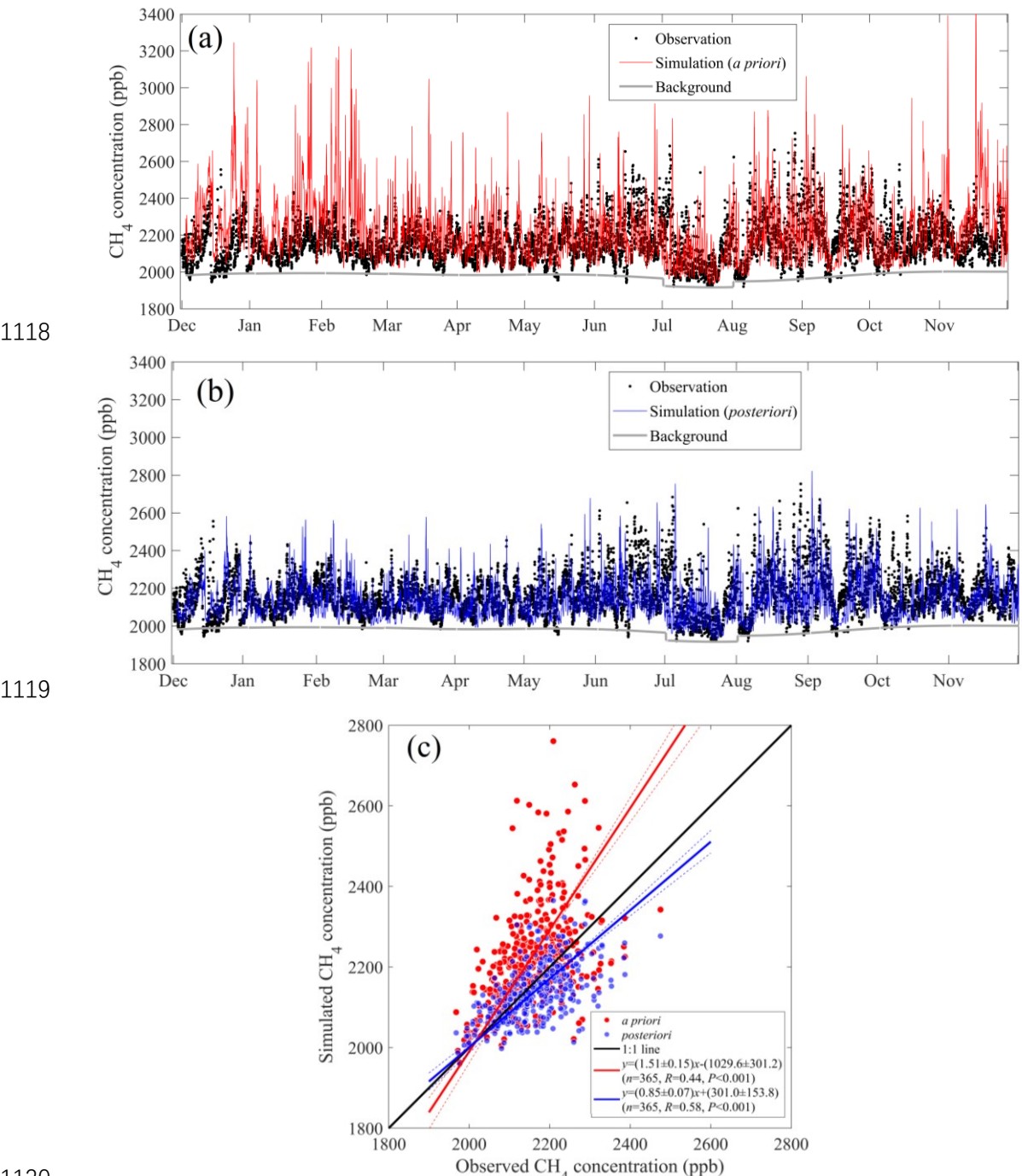



Figure 6. Comparisons of hourly CH$_4$ concentrations at Hangzhou site between observations and
simulations by using (a) *a priori* and (b) *posteriori* emissions, (c) scatter plots of daily CH$_4$
averages by using *a priori* and *posteriori* emissions.







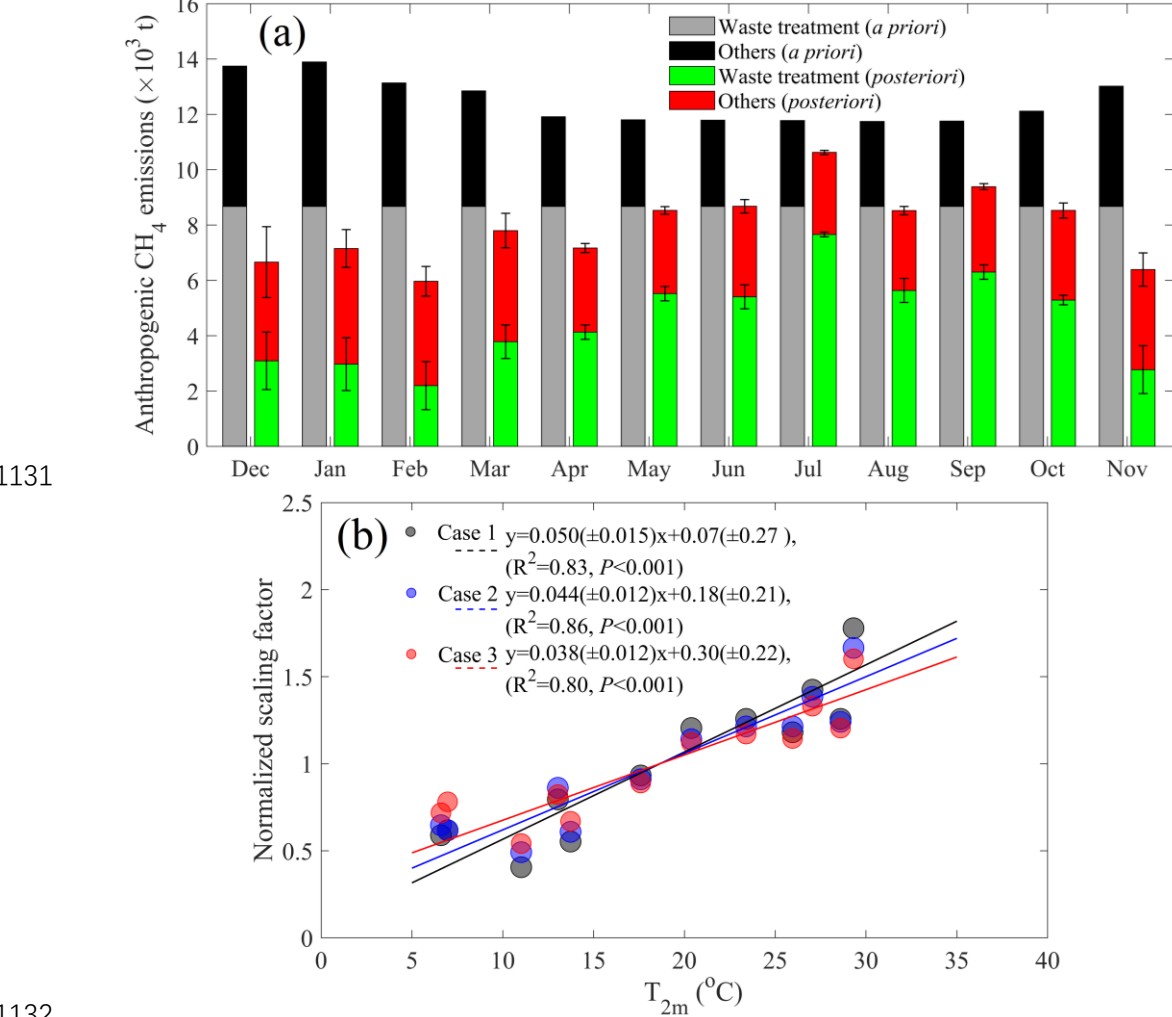

Figure 7. (a) Monthly anthropogenic (excluding agricultural soil) CH$_4$ emissions for *a priori* and *posteriori* emissions for Hangzhou city, (b) relationship between the monthly *posteriori* CH$_4$ emissions and temperature for the three cases discussed in section 2.3 of this text.

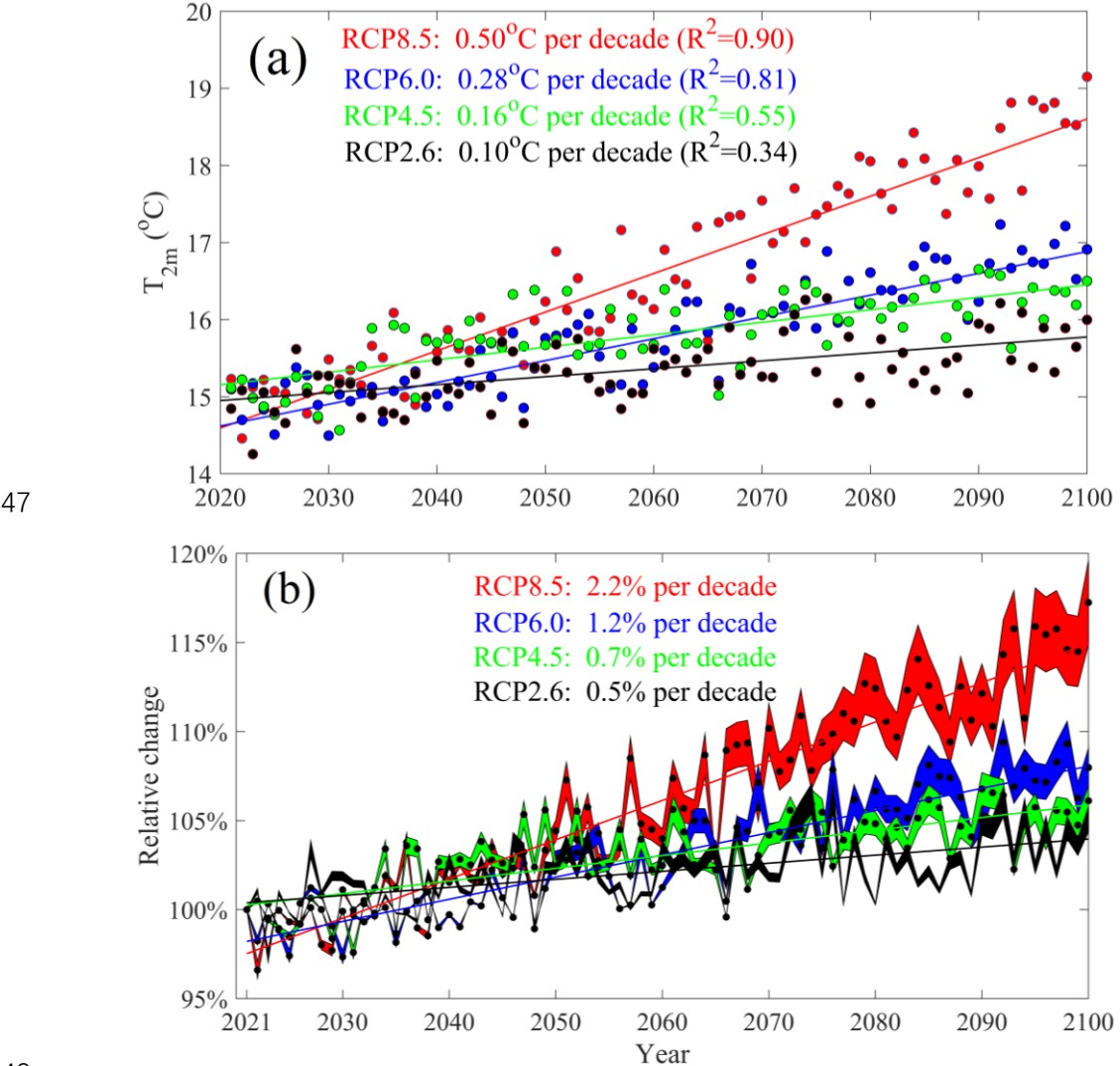

1147

1148

Figure 8. (a) Annual air temperature from year 2021 to 2100 for four different global warming scenarios for Hangzhou city, (b) the projected relative change of waste treatment $CH_4$ emissions (or EFs) for Hangzhou city, note the shading indicates extent of three cases.

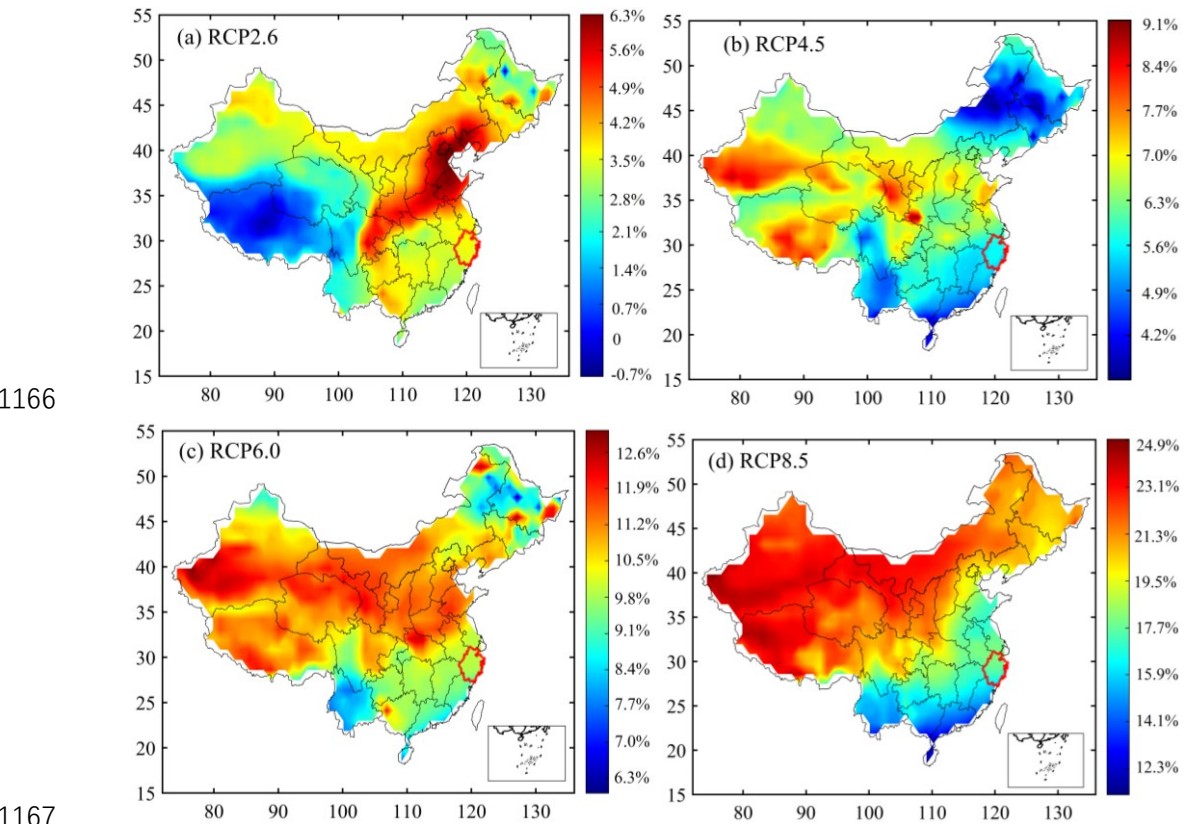

Figure 9. Global warming induced relative changes of waste treatment CH$_4$ EFs by year of 2100 for (a) RCP2.6, (b) RCP4.5, (c) RCP6.0, and (d) RCP8.5 scenarios. Note the red boundary is Zhejiang province.

Table 1. The *posteriori* SFs for different categories in three cases for Hangzhou city, where wetland: natural and agricultural wetland, Waste: waste treatment, PRO: fuel exploitation, RCO: energy for building, Others: the rest anthropogenic emissions. Note Case 1: 3 categories, and 300% uncertainty for waste treatment; Case 2: 5 categories; Case 3: 3 categories, and 200% uncertainty for waste treatment.

| Month | Case 1 | | | Case 2 | | | | | Case 3 | | |
|---|---|---|---|---|---|---|---|---|---|---|---|
| | Wetland | Waste | Others | Wetland | Waste | PRO | RCO | Others | Wetland | Waste | Others |
| 1 | 1.00 | 0.29 | 0.83 | 1.00 | 0.34 | 0.90 | 0.80 | 0.93 | 1.00 | 0.40 | 0.72 |
| 2 | 1.00 | 0.20 | 0.89 | 1.00 | 0.26 | 0.97 | 0.83 | 0.93 | 1.00 | 0.30 | 0.77 |
| 3 | 1.03 | 0.39 | 1.04 | 1.02 | 0.46 | 1.07 | 0.80 | 0.97 | 1.02 | 0.46 | 0.95 |
| 4 | 1.10 | 0.46 | 0.96 | 1.08 | 0.48 | 1.01 | 0.95 | 0.93 | 1.08 | 0.49 | 0.91 |
| 5 | 1.12 | 0.62 | 0.99 | 1.10 | 0.64 | 1.06 | 0.97 | 0.92 | 1.11 | 0.65 | 0.95 |
| 6 | 1.22 | 0.59 | 1.09 | 1.18 | 0.64 | 1.05 | 0.97 | 1.03 | 1.18 | 0.64 | 1.05 |
| 7 | 1.10 | 0.88 | 0.96 | 1.09 | 0.88 | 1.00 | 1.00 | 0.94 | 1.09 | 0.89 | 0.94 |
| 8 | 1.05 | 0.62 | 0.95 | 1.01 | 0.66 | 0.99 | 0.97 | 0.95 | 1.01 | 0.67 | 0.91 |
| 9 | 1.04 | 0.71 | 1.01 | 1.02 | 0.73 | 0.96 | 0.98 | 1.04 | 1.02 | 0.74 | 0.98 |
| 10 | 1.06 | 0.60 | 0.94 | 1.06 | 0.61 | 0.92 | 0.96 | 1.00 | 1.06 | 0.62 | 0.90 |
| 11 | 1.01 | 0.27 | 0.86 | 1.00 | 0.32 | 0.91 | 0.85 | 0.93 | 1.00 | 0.37 | 0.75 |
| 12 | 1.00 | 0.31 | 0.70 | 1.00 | 0.33 | 0.75 | 0.79 | 0.91 | 1.00 | 0.43 | 0.58 |