# Peer review of "Global warming will largely increase waste treatment CH₄ emissions in Chinese Megacities: insight from the first city scale CH₄ concentration observation network in Hangzhou city, China"

_Atmospheric Chemistry and Physics, 2022_

## Author Comment (AC1)

This study uses continuously measured methane measurements at three tower locations in and around Hangzhou, China, to investigate temporal variations of emissions, especially from the treatment of waste. The authors use the WRF-STILT (Weather Research and Forecasting-Stochastic Time-Inverted Lagrangian Transport) model combined with a Bayesian inversion framework to compare the data driven results with the prior emissions inventory. They conclude that emissions have been overestimated for the city of Hangzhou and that there is a seasonality to the emissions that can only be explained by the waste treatment sector.

This topic is very timely and important for understanding the influence of climate change on emissions of this high global warming potential pollutant, but several issues in this paper need clarification before publication.

Thanks so much for these detailed suggestions. All points have been addressed below (review query in Italic; author response in blue). Changes to the text in the manuscript have been marked in blue.

Of the three sites, it appears that only one is in the city (Hangzhou), and one is on a relatively remote mountain (Damingshan). Is the third site, Linan, in a suburb or also background region, as stated on line 177? If this is true, then there is only one site that is truly relevant to determining emissions from the city, since the other two are described as background sites. However, background values are taken from much more remote sites. There can be significant sources between the very remote sites and the urban region being studied, including large cities such as nearby Shanghai between Hangzhou and TAP and RYO, the latter being used almost always as background. The footprint for the Damingshan site is only slightly influenced by emissions in the urban core.

For all three sites used in this study, $CH_4$ concentration at Hangzhou site was used to constrain emissions for Hangzhou city, and $CH_4$ concentration at Linan site was used to constrain emissions for much larger regions as Zhejiang province or Yangtze River Delta Area. The reason of choosing two sites in the emission constraint are mainly based on simulated enhancement contributions from different regions. The explanations are also displayed on lines 399-404 as "We further calculated anthropogenic contributions from Hangzhou city (excluding wetland because of coarser spatial resolution for Hangzhou city) and other provinces, which were 158.4 ppb at Hangzhou site, 30.7 ppb at Linan site, and 10.1 ppb at Damingshan site, respectively. And they accounted for 69.3%, 34.0%, and 16.9% of total anthropogenic enhancements at corresponding sites. These results indicate the $CH_4$ observations at Hangzhou site, which is located at the core urban region, was more influenced by local emissions (mainly for waste treatment and will be discussed later) and contain much higher enhancements than other two sites. The relative contributions from different regions also imply that the observations at Linan and Damingshan sites can present $CH_4$ emissions of much larger region as Zhejiang province or YRD area than Hangzhou city (Figure 4e)."

The reason to use different background sites at the edge of simulation domain instead of Damingshan site have been explained on lines 210-221 as "Note some previous studies of city scale greenhouse gas concentration observation networks chose sites at the edge of urban borders as background in emission inversion system (i.e. Indianapolis, U.S.A., Miles et al., (2017); Los Angeles, U.S.A., Verhulst et al., (2017); Washington, DC-Baltimore, U.S.A., Lopez-Coto et al., (2020); Paris, France, Lian et al., (2021) ), but we chose to use five $CH_4$ background sites as the potential background to be selected including UUM, TAP, YRO, YON and WLG site (Figure 1a), which were much further than the

observations at Damingshan site. This strategy is based on following three reasons: (1) our footprint domain is much larger than Hangzhou city and these five sites are also located close to the edge of model domain; (2) CH$_4$ concentrations within Hangzhou city will be influenced by seasonal varied monsoon and the monthly varied wind directions will lead to obvious changes of CH$_4$ background than only at Damingshan site; (3) our model setups can partition CH$_4$ enhancements from within Hangzhou city and other regions."

[Figure]

Figure 1. (a) WRF-STILT model domain setups, three CH$_4$ concentration observation sites in Hangzhou city, and five CH$_4$ background sites, note the green, red and black dots represent locations for Hangzhou site, Linan site and Damingshan site, respectively, Yangtze River Delta regions is displayed in red boundary, back rectangle represents domain in STILT model, (b) geophysical height within Hangzhou city, (c) land surface categories in Hangzhou city.

Table S1. The choice of CH$_4$ background based on simulated monthly footprint, 'Y' indicates concentration at this background site (or averages of both) will be used as CH$_4$ background for this month.

| Sites | Dec | Jan | Feb | Mar | Apr | May | Jun | Jul | Aug | Sep | Oct | Nov |
|-------|-----|-----|-----|-----|-----|-----|-----|-----|-----|-----|-----|-----|
| TAP   |     |     |     |     |     |     |     |     |     |     |     |     |
| YON   |     |     |     |     |     |     |     | Y   |     |     |     |     |
| RYO   |     | Y   | Y   | Y   | Y   | Y   | Y   | Y   | Y   | Y   | Y   | Y   |
| WLG   |     |     |     |     |     |     |     |     |     |     |     |     |
| UUM   | Y   | Y   |     |     |     |     |     |     |     |     |     |     |

We agree that the simulated enhancements using more remote sites as background contain contributions from other cities as Shanghai, our emission constraint results can also represent Hangzhou city based on the following two reasons: (1) the contribution from Hangzhou city accounted for majority of all enhancement (~70%) for CH$_4$ observations at Hangzhou site; (2) the *posteriori* SFs of the monthly scaling factors mainly represent temporal variations and revised the *a priori* bias, these SFs should be reasonable for a much larger regions and represent the common bias in *a priori* EDGAR inventory.

What emissions did you use for the prior? It seems like you used the EDGAR v6.0 inventory for anthropogenic sources (except rice patties) and WetCHARTs for wetland emissions, including from rice patties. Please state explicitly how you calculated the prior – "a priori" is not mentioned in the WRF-STILT model setup section.

The EDGAR v6.0 and WetCHARTs are used as *a priori* emissions, we revised the sentence as "The most recent inventory of Emission Database for Global Atmospheric Research (EDGAR v6.0), which has 20 categories, and WetCHARTs ensemble mean were used as the *a priori* anthropogenic and natural $CH_4$ emissions." on lines 247-249.

A major assumption of the paper is that waste treatment is the dominant source of emissions and the other anthropogenic sources do not contribute to the seasonality of the observed $CH_4$ measurements. What you show in Figure 4d is that waste treatment contributes most to the $CH_4$ signal, but the other sources are also important. Perhaps you can show a map of the locations of the anthropogenic sources – power plants (especially natural gas powered), landfills, wastewater treatment plants, distribution lines for natural gas, refineries, dairies, rice paddies – especially close to the urban center. Enlarge the urban center to show locations. I am not convinced that you have enough information to discount the influence of other CH4 emissions sources or to characterize the sources in the urban center with only one site, especially when the reader does not know the sources in the region or the general seasonal wind patterns. A measurement that you might consider for the future is ethane, since fossil-fuel-derived CH4 contains measurable C2H6, whereas biological sources (including waste treatment and wetlands) do not. Seasonality due to fossil CH4 is observed in cities, even as far south as Los Angeles. Is rice cultivation seasonal – should you expect some seasonality from this sector?

Here as displayed in Figures 7a, S7, $CH_4$ emissions from waste treatment, and RCO (energy for building) dominated the seasonal variations of $CH_4$ emissions. But contrary to waste treatment, our constrained results indicated the *posteriori* RCO $CH_4$ emissions did not have obvious difference with *a priori* emissions

We added more description on 475-477 as "These *posteriori* SFs for the rest anthropogenic categories and wetland indicated much smaller bias than waste treatment. The monthly *posteriori* SFs for PRO and RCO also illustrated obvious seasonal variations, but were still smaller than the *a priori* seasonality in inventory (Figure S7)."

We double checked the locations of some $CH_4$ emitters as landfills, resident area and RCO (energy for building), they are located in the similar locations as EDGAR, indicating overall good representativity of main anthropogenic $CH_4$ emissions in Hangzhou city. We agree that more tracers as ethane (C2H6) is a good tool to separate $CH_4$ emissions from biological and fossil $CH_4$ emissions, we added this suggestion on lines as "We will use multiple years' $CH_4$ concentration to quantify the influence of new technology and other meteorological variables on waste treatment $CH_4$ emissions in our following study, and we suggest other tracers (i.e. ethane, $^{14}CH_4$) are also important to separate $CH_4$ emissions from biological and fossil $CH_4$ emissions.". on lines 682-686.

[Figure]

Figure 7. (a) Monthly anthropogenic (excluding agricultural soil) CH₄ emissions for *a priori* and *posteriori* emissions for Hangzhou city,

[Figure]

Figure S7. Comparisons of anthropogenic CH₄ emissions between *a priori* and *posteriori* results, PRO: fuel exploitation, RCO: energy for building, the rest anthropogenic emissions: excluding waste treatment, PRO, RCO and agricultural soil.

This paper uses all of the diurnal cycle of the measurements. I definitely agree that emissions at night are not captured if only afternoon measurements are used, as is commonly done. However, one reason most investigations don't use the entire 24-hour record is that WRF does not do a good job with the transport parameters at night, specifically the planetary boundary layer height (PBLH). It is very important to get this right for modeling to produce meaningful results. You don't show how your model performed for this critical parameter. Can you show how the modelled PBLH compares with measurements, even if only a limited number of measurements are available?

We contacted with local meteorological office, but they said there are not available PBLH observations during study period. But we can raise other supporting evidence for the PBLH simulations by citing one of our previous study (Huang et al., 2021), Huang et al. (2021) used the same physical schemes as this study and conducted in Nanjing city from years 2017 to 2018, which is located in the same Domain 2 and vary close to Hangzhou city. Their study found high consistence between observed and simulated PBLH in winter as displayed in following figure.

We cited this reference on lines 440-445 "Note PBLH simulations are important in evaluating model performance, we did not have direct PBLH observations to evaluate model performance, but our previous study used the same physical and PBLH schemes as this study, which was conducted in

Nanjing city in the same Domain 2 and vary close to Hangzhou city. The study found high consistence between observed and simulated PBLH in winter (Huang et al., 2021)."

[Figure]

**Fig. S7**. Diurnal mean planetary boundary-layer height (PBLH).

More detailed comments follow:

Abstract: mention the types of waste included in this study

Thanks so much for pointing it out, we have added "(including solid waste landfills, solid waste incineration and sewage)" on lines 36-37 for clarification.

Line 72: Out of curiosity, what are the top five anthropogenic sources of $CH_4$ in China?

The top five anthropogenic sources in China are fuel exploitation (coal+natural gas+oil), livestock, rice paddies, waste treatment, and household use, the top four (fuel exploitation, livestock, rice paddies and waste treatment) of them accounted nearly 90% of national total anthropogenic $CH_4$ emissions.

Line 91: USEPA

Thanks for pointing out this typo, we have changed "USPA" with "USEPA".

Line 106: replace "absence" with "omission"

Done as suggested, we have replaced it with "omission".

Lines 143-145: City-scale studies have not focused on waste treatment sources because there are many sources, as in Hangzhou. Yadav et al. (2019; JGR Atmospheres) were able to see the effects of the closure of a landfill in the Los Angeles, CA area that was included in the prior inventory and not seen in the modelled results.

Done as suggested, we have revised this sentence as "And to our best knowledge, there is few tower-based observation inversion studies which focuses on waste treatment emissions at city scale or much larger regional scales especially in China. Only one study in Los Angeles, U.S.A. used tower-based $CH_4$ concentration and found the influence of landfill site closure on $CH_4$ emissions, which was not included in *a priori* inventory (Yadav et al., 2019)." on lines 149-153.

Yadav, V., Duren, R., Mueller, K., Verhulst, K. R., Nehrkorn, T., and Kim, Jet., Spatio-temporally resolved methane fluxes fromthe Los Angeles megacity J. Geophys. Res. Atmos. 124, 5131–5148 (2019).

Pages 6-7: In the description of the sites, please summarize the regional, seasonal wind patterns and any differences between the sites.

Done as suggested, we added more description for clarification as "Based on the wind direction for three sites, there are not obvious difference of seasonal wind direction patterns among them. The prevailing wind direction from October to February was from the north, which changed to east from February to May and then changed to south during the monsoon in summer." on lines 189-192.

Lines 188-190: How frequently were standards run? What uncertainty, including both precision and accuracy, did you assign for the measurements?

The analyzer was calibrated every 6 hours with the measured precision and accuracy within 2 ppb and 1 ppb, respectively. These uncertainties were pretty small when compared with background and we have considered these uncertainties in the Bayesian inversion. We revised it on lines 198-200 as "two different standard gas was measured every 6 hours and a linear two-point fit was used to calibrate observations, with the precision and accuracy of 2 ppb and 1 ppb.".

Line 238: What does "fuel exploitation from coal, oil, and natural gas" include? Extraction, transportation, refining, distribution, and combustion, or some subset of these?

The fuel exploitation from coal, oil, and natural gas in EDGAR contains all related processes as extraction, transportation, refining, distribution as list in IPCC database (https://www.ipcc-nggip.iges.or.jp/EFDB/find_ef.php). But such category is minimum in our study region when compared with other sources. We revised it as "PRO (all processes related to fuel exploitation from coal, oil, and natural gas)" on lines 258.

Line 239: How and where is the energy for buildings generated? E.g., natural gas power plants in the suburbs, coal burned in the buildings, …?

Here in Hangzhou, we think the energy for buildings mainly contains nature gas escape from household use. we revised this sentence as "RCO (energy for buildings, mainly containing nature gas escape from household use)" on lines 259-260 for clarification.

Lines 241-245: Is 0.5° high enough spatial resolution for your study region?

Here considering the WetCHARTs simulations have been widely used as $CH_4$ emissions from wetland in many previous studies, and wetland $CH_4$ emissions was pretty small compared with other $CH_4$ emissions. Hence we decided to use 0.5° WetCHARTs emissions。

Line 287: reference for CCGCRV? Thoning et al., 1989, JGR 94, 8549-8565; https://gml.noaa.gov/ccgg/mbl/crvfit/crvfit.html

Done as suggested, we added this reference.

Thoning, K. W., Tans, P. P., and Komhyr, W. D.: Atmospheric carbon dioxide at Mauna Loa observatory 2. Analysis of the NOAA/GMCC data, 1974–1985, J. Geophys. Res.-Atmos., 94, 8549–8565, https://doi.org/10.1029/JD094iD06p08549, 1989.

Line 295: Is it meaningful to give an annual average when 1-2 months are missing data?

Done as suggested, we added the calculated uncertainty and revised it as "(annual mean of 2013.4±(3) ppb, where the uncertainty is calculated when assuming the missing data in September and October varied between August and November)," on lines 321-322.

Line 295: replace "variations" with "trends"
Done as suggested, we replaced "variations" with "trends".

Line 296: What are the "similar atmospheric transport processes?" Summarize seasonal wind direction and speed patterns.
Considering the wind speed at different height should change largely, here the "similar atmospheric transport processes" mainly means the wind directions and synoptic process (i.e. monsoon). We have added more clarification with "such as synoptic process (i.e. monsoon) and seasonal changing wind direction as summarized above." On lines 324-325.

Line 309: replace "YON" with "TAP"
Done as suggested and thanks so much for catching up this typo.

Line 310: replace "temporal" with "spatial"
Done as suggested.

Lines 320-321: Figure 3 does not show significant differences in the size of the footprints at the different sites. You might want to expand the scale to show this.
Done as suggested, we have expanded the scale as displayed below.

[Figure]

Lines 323-326: Cities shown significant diurnal variation in PBLH.
Done as suggested, we revised this sentence as "and it will show significant diurnal variation in PBLH, especially have higher nighttime PBLH"

Line 345: Not sure what you mean by "amplitudes" here – amplitude of the seasonal variations? I don't see obvious differences. The absolute average abundances are different.
Done as suggested, we changed "amplitudes" with "relative variations".

Line 348: The simulated data for Linan actually approximate the observations very well!
Done as suggested, we changed this sentence as "We found the simulations at Linan site shows overall good agreement with observation, but still with slight overestimation from January to April and underestimation from May to September." on lines 383-384.

Line 364: It is very much to be expected that the Hangzhou site is more influenced by local emissions since it is in the urban core. What are the major emitters within 5-10 km of the site?

It seems the major emissions within 5-10 km of Hangzhou site are waste treatments. We revised this sentence as "was more influenced by local emissions (mainly for waste treatment and will be discussed later)" on lines 400-401.

Lines 366-368: How did you show that the Linan and Damingshan sites are influenced by emissions from a much larger region? The footprints don't indicate this.

The simulated $CH_4$ enhancement contributions from Hangzhou city was the highest (69.3%) than Lian (34.0%) and Damingshan (13.0%), indicating the rest contributions are from much further regions.

We revised this sentence as "The relative contributions from different regions also imply that the observations at Linan and Damingshan sites can present $CH_4$ emissions of much larger region as Zhejiang province or YRD area than Hangzhou city (Figure 4e)." on lines 402-404 for more clarification.

Lines 375-378: Can you give a reference for the statement that waste treatment emissions are larger during the daytime than at night?

Here we mean the waste treatment $CH_4$ emissions are sensitive to temperature, where the large diurnal variations and seasonal variations of temperature can lead to large variations of its emissions. We double checked the references and have not found the direct diurnal observations as eddy covariance. Hence, we deleted "are larger during the daytime than at night" and revised this sentence as "which should be highly sensitive to temperature and indicates obvious diurnal and seasonal patterns (Mønster et al., 2019; Kumar et al., 2022)." on lines 412-414.

Line 420: Emissions from waste treatment dominated the total $CH_4$ and the seasonal pattern, but you do show significant seasonal variations for the other anthropogenic sources in Figure 7a. Can you split those up at all? Can you say anything about the natural gas distribution infrastructure – more leaks in winter than summer, …?

The monthly variations of *a priori* EDGAR emissions was mainly driven by RCO (Energy for buildings), which changed from the highest 22% in winter to lowest ~8% in summer. Such information indicates the *a priori* inventory assigned more leaks from natural gas distribution infrastructure in winter than in summer.

To make clarification, we added "The seasonality in *a priori* EDGAR inventory was mainly dominated by RCO (Energy for buildings), with proportions to total anthropogenic emissions changed from the highest 22% in winter to lowest ~8% in summer. Such information indicates the *a priori* inventory assigned more leaks from natural gas distribution infrastructure in winter than in summer." on lines 459-462.

[Figure]

Figure S7. Comparisons of anthropogenic CH₄ emissions between *a priori* and *posteriori* results, PRO: fuel exploitation, RCO: energy for building, the rest anthropogenic emissions: excluding waste treatment, PRO, RCO and agricultural soil.

we also added one more figure (Figure S7) in supplementary file, and added more discussions as "Besides, the annual mean *posteriori* SFs varied between 0.87 and 0.94 for rest total anthropogenic categories (excluding agricultural soil), and were 0.97 for PRO (fuel exploitation) and 0.91 for RCO (energy for building), respectively; the annual mean *posteriori* SFs and were 1.05 and 1.05 for wetland (including agricultural soil and natural wetland). These *posteriori* SFs for the rest anthropogenic categories and wetland indicated much smaller bias than waste treatment. The monthly *posteriori* SFs for PRO and RCO also illustrated obvious seasonal variations, but were still smaller than the *a priori* seasonality in inventory (Figure S7)." on lines 441-447.

Line 479: Where are these values of SFs shown? They are not from Table S2.
We have mentioned on line 470 as "The derived monthly *posteriori* SFs for each emission source were displayed in Table 1 for Hangzhou city.".

Figures: In general, please improve the resolution of the figures. It is very difficult to impossible to read the small text, even when expanding the figures on the screen.
Done as suggested, the main reason why the resolution of all figures seems low is caused when MS in world version transformed in pdf version, and we have updated the new pdf version with high resolutions.

Figure 1: What are the divisions within Hangzhou City?
The divisions in Hangzhou city are different districts, we have added it in the caption of Figure 1.

Figure 2: Use the same color schemes on all figures and parts within figures for the same sites.
Done as suggested, we have used the same color schemes on all figures as displayed below.

[Figure]

Figure 3: Replace "lg" with "log." Are the waste treatment CH4 emissions in panel (e) also from EDGAR v6.0?

Here both footprint and $CH_4$ emissions are expressed using log10, hence we added lg in these figures.

The waste treatment $CH_4$ emissions in panel (e) is from EDGAR v6.0, we have added the data sources on the caption of Figure 3e on line 977.

Figure 4: Need higher resolution graphics, especially for panel (d) and (e). The note at the end of the caption may be incorrect. In panel (e), is it true that the blue color for the bar charts include all of Zhejiang, including Hangzhou? Do the blue regions in the pie charts represent Zhejiang minus Hangzhou?

Done as suggested, we increased the size and resolution of Figure 4.

Yes, in panel (e), the blue color for the bar charts include all of Zhejiang, including Hangzhou; and the blue regions in the pie charts represent the rest region of "Zhejiang minus Hangzhou". We have added more descriptions for clarification as "Note the blue color for the bar charts include all contributions from "Zhejiang", including "Hangzhou"; and the blue regions in the pie charts represent rest regions of "Zhejiang minus Hangzhou" on lines 1003-1005.

Figure 8: What region is this figure describing?

Thanks so much for pointing it out, this figure is for Hangzhou city, and we have added "for Hangzhou city" at the caption of Figure 8.

---

## Author Comment (AC2)

Summary:

Review of "Global warming will largely increase CH$_4$ emissions from waste treatment: insight from the first city scale CH$_4$ concentration observation network in Hangzhou city, China" by Hu et al. 2022 for Atmospheric Chemistry and Physics.

Hu et al. use atmospheric observations and modelling tools (lagrangian) to estimate methane emissions from an important Megacity. Relying on EDGAR V6.0 they analyse the sectorial contribution to atmospheric CH$_4$ enhancements and then optimize fluxes using a Bayesian framework. The results indicate an overestimate of local emissions by EDGAR V6.0. The seasonal bias between a priori and a posteriori fluxes is attribute to waste sector emissions and a temperature sensitivity is calculated. Using IPCC scenarios the authors than quantify the temperature-specific component of the waste sector emission factor changes for the coming decades.

Overall, the paper is clear and can be followed easily. However, the study lacks some critical assessments around the choice of EDGAR V6.0 and the implications of that choice. Furthermore, the study should be clearer on the fact that the suggested effect could be fully compensated by other parameters affecting the waste sector emission factor. It also would be useful to specify that a single city study should not be scaled globally, but that is surely has an important message for CH$_4$ emissions in Chinese Megacities. Given the importance of this region for future climate change this study is surely of interest to the wider scientific community and especially ACP readers. After addressing the general and specific comments this manuscript would appear suitable for publication.

Thanks so much for these detailed suggestions and we have made extensive revisions based on these comments.

General comments:

The title implies a global impact; however, it only provides results for one urban region. Also, country-specific waste management strategies (e.g. highly localized waste separation stations) call into question how much the results from this region can be extrapolated beyond Chinese Megacities.

Done as suggested, the global warming will lead temperature increase in China and most part of other countries. Here we only quantified the temperature sensitivity of waste treatment CH$_4$ emissions, and this sensitivity can be also used both for China and other countries, especially for urban areas.

We finally changed the title as ""Global warming will largely increase waste treatment CH$_4$ emissions in Chinese Megacities: insight from the first city scale CH$_4$ concentration observation network in Hangzhou city, China"

Besides, we added more discussion to make clarification as "Considering the temperature sensitivity of waste treatment CH$_4$ EFs are caused by microbial process at the regional scales, it can represent general conditions of different cities or landfills." on lines 551-553.

This study only assess the influence of temperature on the emission factor for waste although previous work has shown the importance of other meteorological parameters such as atmospheric pressure changes, water content and management strategies. It is unclear that local climate change would not also affect these parameters as well. This could reduce or strengthen the suggest increase in emissions. The

authors also do not discuss how relevant temperature is as a parameter when compared to the others mentioned above.

[Figure]

Figure S8. (a) Relationship between monthly averaged atmospheric pressure and normalized SFs, (b) relationship between monthly averaged atmospheric pressure and $T_{2m}$, and (c) relationship between monthly precipitation and $T_{2m}$

Done as suggested, we analyzed the relationship between monthly averaged atmospheric pressure and normalized SFs (Figure S8a), and the relationship between atmospheric pressure and $T_{2m}$ (Figure S8b), and relationship between precipitation and $T_{2m}$ (Figure S8c). They displayed positive linear relationship between precipitation and $T_{2m}$, and negative linear relationship between monthly averaged atmospheric pressure and normalized SFs, and between atmospheric pressure and $T_{2m}$. Considering air temperature always displays negative relationship with atmospheric pressure as warmer air temperature, lighter air mas and lower atmospheric pressure in summer, and colder air temperature, heavier air mass and higher atmospheric pressure in winter. Hence, the temperature can be used to represent co-influence of both temperature and atmospheric pressure, and we only focus on the influence of temperature on $CH_4$ emissions and will add more supporting data in following research.

We added this figures in Supplementary file and also added more clarification on lines 549-562 as "We should note the precipitation, soil water content and atmospheric pressure can also have obvious influence

on CH$_4$ emissions, and considering the fact that we have not conducted field measurement in landfills and landfills are usually covered by metal or plastic in China to avoid the spread of odor smell, hence reanalysis data cannot represent real soil water contents in these site scale landfills. Precipitation and atmospheric pressure showed obvious linear relationship with temperature as displayed in Figure S8. They displayed positive linear relationship between precipitation (affect water content) and T$_{2m}$, and negative linear relationship between monthly averaged atmospheric pressure and T$_{2m}$. We also found negative relationship between atmospheric pressure and normalized SFs (Figure S8a). Considering air temperature always displays negative relationship with atmospheric pressure as warmer air temperature coincides with lighter air mas and lower atmospheric in summer, and colder air temperature coincides with heavier air mass and higher atmospheric pressure in winter. Hence, the temperature can be used to represent co-influence of both temperature and atmospheric pressure, and we only focus on the influence of temperature on CH$_4$ emissions and will add more supporting data in following studies.".

As replied in details below, we answered why only using temperature in this study. To make clarification, we added "We should note that new technology and other meteorological variables can also influence waste treatment CH$_4$ emissions. The reason to only use temperature in this study is mainly for the reason that we only constrained the emission at monthly scale in one year, and derived twelve datasets of *posteriori* CH$_4$ emissions. Besides, temperature is considered as the main factor in controlling monthly and annual variations of waste treatment CH$_4$ emissions, and can be used to represent co-influence of other meteorological parameters as atmospheric pressure. We will use multiple years' CH$_4$ concentration to quantify the influence of new technology and other meteorological variables on waste treatment CH$_4$ emissions in our following study, and we suggest other tracers (i.e. ethane, $^{14}$CH$_4$) are also important to separate CH$_4$ emissions from biological and fossil CH$_4$ emissions." on lines 677-686.

This study uses EDGAR CH4 without critically assessing its limitations. EDGAR is coarse resolution 0.1x0.1 degree for urban studies and was shown to have biases in some high-density urban areas. e.g. Vogel et al. 2012 (https://doi.org/10.1080/1943815X.2012.691884). Why do you rely solely on EDGAR and why do you believe its spatial disaggregation to be correct?

Thanks so much for this suggestion, the reason to choose EDGAR is that (1) We agree that there are many CH$_4$ inventories for other developed regions and countries (i.e. France, U.S.A., Germany) with high resolutions, but for all available CH$_4$ inventories that covered China, the spatial resolution of EDGAR ($0.1°×0.1°$) is the highest, and the update date for EDGAR is most to date; (2) most of previous studies that constrain emissions by atmospheric inversion studies have chosen EDGAR, and our results can be used to compare with previous studies; (3) our preliminary simulation of CH$_4$ concentrations showed generally good performance with observations, indicating its spatial distributions in Hangzhou city can be with relatively small bias even with potential large bias for magnitude, which will be constrained by our inversion method. We will apply more inventories in the following study by using multiple years' CH$_4$ observations as noted in this MS.

To make clarifications, we added "We should note there are many CH$_4$ inventories for some developed regions and countries (i.e. France, U.S.A., Germany) with high spatial resolutions, the reasons to choose EDGAR as *a priori* anthropogenic emissions are: (1) for all available CH$_4$ inventories that covered China, the spatial resolution of EDGAR ($0.1°×0.1°$) is the highest, and it provide most up-to date results; (2) most of previous studies that constrain emissions by atmospheric inversion studies also chose EDGAR, and our

results can be directly compared with previous studies; (3) the preliminary simulation of $CH_4$ concentrations showed generally good performance with observations, indicating its spatial distributions in Hangzhou city has relatively small bias even with potential large bias for magnitude, which will be constrained by our atmospheric inversion method." on lines 246-255.

Specific comments:

Line 36 and line 75:

Please provide a source for the claim that waste emissions contribute over 50% of $CH_4$ emissions at city-scale. For which cities and regions does this apply?

Done as suggested, we revised this sentence as "Furthermore, its contribution is even larger than 50% at city scale especially for megacities, where both active and closed household waste (including landfills and waste water systems) are located and found as super emitters (Williams et al., 2022; Maasakkers et al., 2022). A large number of Chinese landfills were mainly constructed at the suburban more than 5-10 years ago, and with the urban area expanding in recent decades, the locations of many landfills are now in urban scope (Zhejiang Statistical Yearbook 2018-2019). Besides, the decreasing area of agricultural sector (rice paddies and husbandry) in megacities also makes their emissions ignorable when compared with waste treatment." on lines 74-81. The reference was Maasakkers et al. (2022) and we added this refence here.

Also please provide evidence that most household waste is located in cities and not in landfills outside the cities. In some regions landfills are located outside the city limits.

Done as suggested. As surveyed for local conditions in Hangzhou city and some typical Chinese cities, the landfills were mainly constructed at the suburban more than 5-10 years ago, and with the urban area expanding in recent decades, the locations of many landfills are now in urban regions.

We added "A large number of Chinese landfills were mainly constructed at the suburban more than 5-10 years ago, and with the urban area expanding in recent decades, the locations of many landfills are now in urban regions (Zhejiang Statistical Yearbook 2018-2019)." on lines 77-79.

Zhejiang Provincial Bureau of Statistics, Survey Office of the National Bureau of Statistics in Zhejiang, Zhejiang Statistical Yearbook 2018-2019 (China Statistics Press, Beijing, China, 2019)

Line 75: Please add a critical discussion of the importance of active and closed landfills, waste water systems and household waste in residential areas. Recent work has shown that waste water can be a significant source at urban scale. E.g. Williams et al. 2022 (https://doi.org/10.1021/acs.est.2c06254).

Done as suggested, we added more discussion as "Furthermore, its contribution is even larger than 50% at city scale especially for megacities, where both active and closed household waste (including landfills and waste water systems) are located and found as super emitters (Williams et al., 2022; Maasakkers et al., 2022)." on lines 74-77.

Williams, J. P., Ars, S., Vogel, F., Regehr, A., & Kang, M. (2022). Differentiating and Mitigating Methane Emissions from Fugitive Leaks from Natural Gas Distribution, Historic Landfills, and

Manholes in Montréal, Canada. Environmental Science & Technology. https://doi.org/10.1021/acs.est.2c06254

Line 79-83: this review fails to mention the critical impact of atmospheric pressure changes on emissions. As shown by e.g. Kissas et al. 2022 (https://www.sciencedirect.com/science/article/pii/S0956053X21006310) and references therein. Emissions can be increased by orders of magnitude due to this effect.

Thanks so much for pointing it out, we added "atmospheric pressure" here, and citied this reference of Kissas et al. (2022).
Kissas K , Ibrom A , Kjeldsen P , et al. Methane emission dynamics from a Danish landfill: The effect of changes in barometric pressure. Waste Management, 2022, 138:234-242.

Line 137: Given the strong influence from barometric pressure on landfill CH4 emissions it is critical to discuss the clear-sky bias of satellites here. Satellite observations are too sparse to be up-scaled to estimate annual totals.

Done as suggested, we added "Given the strong influence from atmospheric pressure on landfill $CH_4$ emissions, satellite observations are too sparse to be up-scaled to estimate annual total because satellite observations are almost conducted in clear-sky conditions and cannot represent atmospheric pressure and $CH_4$ emissions in cloudy or rainy days." on lines 139-142.

Line 165: The described study can only assess the temperature component of the EF changes but neglects pressure changes as well as all the other factors outlined in line 79-83, e.g. water content oxidation efficiency, landfill gas collection.

As answered below for related questions, to make clarification, we added "We should note the precipitation, soil water content and atmospheric pressure can also have obvious influence on $CH_4$ emissions, and considering the fact that we have not conducted field measurement in landfills and landfills are usually covered by metal or plastic in China to avoid the spread of odor smell, hence reanalysis data cannot represent real soil water contents in these site scale landfills. Precipitation and atmospheric pressure showed obvious linear relationship with temperature as displayed in Figure S8. They displayed positive linear relationship between precipitation (affect water content) and $T_{2m}$, and negative linear relationship between monthly averaged atmospheric pressure and $T_{2m}$. We also found negative relationship between atmospheric pressure and normalized SFs (Figure S8a). Considering air temperature always displays negative relationship with atmospheric pressure as warmer air temperature coincides with lighter air mas and lower atmospheric in summer, and colder air temperature coincides with heavier air mass and higher atmospheric pressure in winter. Hence, the temperature can be used to represent co-influence of both temperature and atmospheric pressure, and we only focus on the influence of temperature on $CH_4$ emissions and will add more supporting data in following studies." on lines 549-562.

And also added "We should note that new technology and other meteorological variables can also influence waste treatment $CH_4$ emissions. The main reason to only use temperature in this study is that we only constrained the emissions at monthly scale in one year, and derived twelve datasets of *posteriori* $CH_4$ emissions. Besides, temperature is considered as the main factor in controlling monthly and annual variations of waste treatment $CH_4$ emissions, and can be used to represent co-influence of other

meteorological parameters as atmospheric pressure. We will use multiple years' $CH_4$ concentration to quantify the influence of new technology and other meteorological variables on waste treatment $CH_4$ emissions in our following study, and we suggest other tracers (i.e. ethane, $^{14}CH_4$) are also important to separate $CH_4$ emissions from biological and fossil $CH_4$ emissions." on lines 677-686.

Line 273-282: How where these prior uncertainties calculated/determined? They seem to strongly differ from Solazzo et al. 2021 (https://doi.org/10.5194/acp-21-5655-2021)
Here in Solazzo et al. 2021 (https://doi.org/10.5194/acp-21-5655-2021)
, the uncertainty of $CH_4$ from waste treatment was 30%~50%, which was calculated mainly from activity data and EFs at the country scale, we should note many previous studies also found the uncertainty will largely increase with study region decrease, and also as stated on lines 118-120 "A recent study by comparing waste treatment $CH_4$ emissions among different inventories also reported that the EDGAR v5.0 and CEDS (Community Emissions Data System) inventories were 21~153% higher than other inventories", and "There was only one recent study by using satellite observations and focused on urban waste treatment $CH_4$ emissions, it found annual $CH_4$ emissions from four cities were 1.4 to 2.6 times larger than inventories in India and Pakistan," we finally choose to assign the larger uncertainty to better constrain $CH_4$ emissions. Furthermore, As found in this study for figure 7a, our research found the *a priori* monthly $CH_4$ emissions from waste treatment were 1.5-3 times of *posteriori* emissions.

To make clarification, we added "Although previous study derived uncertainty of $CH_4$ from waste treatment and other categories, which varied between 30% and 50%, these uncertainties were calculated mainly from activity data and EFs at the country scale on annual average (Solazzo et al. 2021). We should also note $CH_4$ emission uncertainty will largely increase with study region decreasing, as stated above the relative difference among different inventories can reach to 150%. Considering the disaggregation of spatial distributions and temporal variations, $CH_4$ emission uncertainties can be much larger at urban and monthly scales." on lines 295-301.

Solazzo, E., Crippa, M., Guizzardi, D., Muntean, M., Choulga, M., and Janssens-Maenhout, G.: Uncertainties in the Emissions Database for Global Atmospheric Research (EDGAR) emission inventory of greenhouse gases, Atmos. Chem. Phys., 21, 5655–5683, https://doi.org/10.5194/acp-21-5655-2021, 2021.

Line 287: Please provide a reference for the CCGCRV fitting method.
Done as suggested, we added the reference of "Thoning et al., 1989".

Thoning, K. W., Tans, P. P., and Komhyr, W. D.: Atmospheric carbon dioxide at Mauna Loa observatory 2. Analysis of the NOAA/GMCC data, 1974–1985, J. Geophys. Res.-Atmos., 94, 8549–8565, https://doi.org/10.1029/JD094iD06p08549, 1989.

Line 336: Please provide a reference for the emissions from waste separation stations.
Here we want to express the idea that besides the large waste landfills located in some special locations, the building of high density of waste separation stations will also potentially lead to $CH_4$ emissions, and we have added the reference which just mentioned the building of high density of waste separation

stations (Tian et al., 2022). but to our best knowledge, we have not found related studies that point out these $CH_4$ sources.

Tian, J., Gong, Y., Li, Y., Chen, X., Zhang, L., & Sun, Y. (2022). Can policy implementation increase public waste sorting behavior? The comparison between regions with and without waste sorting policy implementation in China. Journal of Cleaner Production, 132401.

Line 344: Please quantify the consistency of the temporal patterns by providing Pearson's r values for all time series shown in Figure 4.

Done as suggested, we added corresponding statistic data as "The mean bias (MB), root mean squared error (RMSE), and correlation coefficient (R) between daily observations and a priori simulations were 64.1 ppb, 129.2 ppb and 0.44, respectively, for Hangzhou site; and were -6.0 ppb, 57.1 ppb, 0.50 for Linan site, 36.2 ppb, 55.6 ppb, 0.54 for Damingshan site." on lines 380-383.

Line 357: The finding that waste dominates emissions here strongly relies on the spatial patterns of EDGER being correct also previous work has shown limitations of EDGAR to capture CH4 emission patterns in urban areas, see e.g. Pak et al. 2021 (https://doi.org/10.1016/j.atmosenv.2021.118319)

Here considering the fact that locations of landfills, which is the largest anthropogenic $CH_4$ emitter in Hangzhou city, are very close to the core urban area, hence we believe the spatial patterns of EDGAR in study region can be with much less bias as stated in above mentioned reference.

We also added more explanation to make clarification as "Although a few previous studies found limitations of EDGAR inventory to capture $CH_4$ emission patterns in some urban areas (Pak et al., 2021), here considering the fact that locations of landfills, which is the largest anthropogenic $CH_4$ emitter in Hangzhou city, are very close to the core urban area and in high consistence with EDGAR, hence we believe the spatial patterns of EDGAR in study region can be reliable.". on lines 363-368.

Pak N M , Heerah S , Zhang J , et al. The Facility Level and Area Methane Emissions inventory for the Greater Toronto Area (FLAME-GTA)[J]. Atmospheric Environment, 2021, 252(9):118319.

Line 424: How much do daytime and all-day average concentrations differ at the Hangzhou site?

The annual averages of daytime and all-day average concentrations were 2112.4 and 2156.0 ppb at Hangzhou site, respectively, and more comparisons between daytime and all-day average concentrations are displayed in Figure 5 for three sites. We added "The annual averages of daytime and all-day average concentrations were 2112.4 and 2156.0 ppb at Hangzhou site, respectively, and more comparisons between daytime and all-day average concentrations are displayed in Figure 5 for three sites." on lines 465-468.

Line 425: Here you are assuming strong changes in waste-related methane emissions, without any references, while EDGAR V6.0, which you used as a prior assumes constant emissions.

Done as suggested, here we first added more clarification on lines 413-416 as "Here as concluded above that the main $CH_4$ component in Hangzhou city was waste treatment (Figure 3f), which should be highly sensitive to temperature and indicates obvious diurnal and seasonal patterns (Mønster et al., 2019; Kumar et al., 2022).".

And then revised the sentence on line 469 as "which have much smaller diurnal variations than waste treatment as stated above (Mønster et al., 2019; Kumar et al., 2022)."

Line 482: Your study nicely shows the temporal bias of EDGAR V6.0, what about a potential spatial or sectorial bias?
Thanks so much for this positive comment, for the spatial bias and considering there only two sites to be used, we only showed that "and the annual anthropogenic $CH_4$ emissions were largely overestimated by 36.0% in Hangzhou city but underestimated by 7.0% in the larger region of the Zhejiang Province or YRD area."

And the sectorial bias has already be list in Table 1 and Figure 7 for main categories, we also added one more figure (Figure S7) in supplementary file, and added more discussions as "Besides, the annual mean *posteriori* SFs varied between 0.87 and 0.94 for rest total anthropogenic categories (excluding agricultural soil), and were 0.97 for PRO (fuel exploitation) and 0.91 for RCO (energy for building), respectively; the annual mean *posteriori* SFs and were 1.05 and 1.05 for wetland (including agricultural soil and natural wetland). These *posteriori* SFs for the rest anthropogenic categories and wetland indicated much smaller bias than waste treatment. The monthly *posteriori* SFs for PRO and RCO also illustrated obvious seasonal variations, but were still smaller than the *a priori* seasonality in inventory (Figure S7)." on lines 477-483.

Table 1. The *posteriori* SFs for different categories in three cases, where wetland: natural and agricultural wetland, Waste: waste treatment, PRO: fuel exploitation, RCO: energy for building, Others: the rest anthropogenic emissions.

| Month | Case 1 | | | Case 2 | | | | | Case 3 | | |
|---|---|---|---|---|---|---|---|---|---|---|---|
| | Wetland | Waste | Others | Wetland | Waste | PRO | RCO | Others | Wetland | Waste | Others |
| 1 | 1.00 | 0.29 | 0.83 | 1.00 | 0.34 | 0.90 | 0.80 | 0.93 | 1.00 | 0.40 | 0.72 |
| 2 | 1.00 | 0.20 | 0.89 | 1.00 | 0.26 | 0.97 | 0.83 | 0.93 | 1.00 | 0.30 | 0.77 |
| 3 | 1.03 | 0.39 | 1.04 | 1.02 | 0.46 | 1.07 | 0.80 | 0.97 | 1.02 | 0.46 | 0.95 |
| 4 | 1.10 | 0.46 | 0.96 | 1.08 | 0.48 | 1.01 | 0.95 | 0.93 | 1.08 | 0.49 | 0.91 |
| 5 | 1.12 | 0.62 | 0.99 | 1.10 | 0.64 | 1.06 | 0.97 | 0.92 | 1.11 | 0.65 | 0.95 |
| 6 | 1.22 | 0.59 | 1.09 | 1.18 | 0.64 | 1.05 | 0.97 | 1.03 | 1.18 | 0.64 | 1.05 |
| 7 | 1.10 | 0.88 | 0.96 | 1.09 | 0.88 | 1.00 | 1.00 | 0.94 | 1.09 | 0.89 | 0.94 |
| 8 | 1.05 | 0.62 | 0.95 | 1.01 | 0.66 | 0.99 | 0.97 | 0.95 | 1.01 | 0.67 | 0.91 |
| 9 | 1.04 | 0.71 | 1.01 | 1.02 | 0.73 | 0.96 | 0.98 | 1.04 | 1.02 | 0.74 | 0.98 |
| 10 | 1.06 | 0.60 | 0.94 | 1.06 | 0.61 | 0.92 | 0.96 | 1.00 | 1.06 | 0.62 | 0.90 |
| 11 | 1.01 | 0.27 | 0.86 | 1.00 | 0.32 | 0.91 | 0.85 | 0.93 | 1.00 | 0.37 | 0.75 |
| 12 | 1.00 | 0.31 | 0.70 | 1.00 | 0.33 | 0.75 | 0.79 | 0.91 | 1.00 | 0.43 | 0.58 |

[Figure]

Figure S7. Comparisons of anthropogenic CH$_4$ emissions between *a priori* and *posteriori* results, PRO: fuel exploitation, RCO: energy for building, the rest anthropogenic emissions: excluding waste treatment, PRO, RCO and agricultural soil.

Line 493-496: Have you investigated the correlation of monthly CH$_4$ emission changes with soil water content, precipitation or other parameters you listed in line 79-83?

Done as suggested, we added "We should note the precipitation, soil water content and atmospheric pressure can also have obvious influence on CH$_4$ emissions, and considering the fact that we have not conducted field measurement in landfills and landfills are usually covered by metal or plastic in China to avoid the spread of odor smell, hence reanalysis data cannot represent real soil water contents in these site scale landfills. Precipitation and atmospheric pressure showed obvious linear relationship with temperature as displayed in Figure S8. They displayed positive linear relationship between precipitation (affect water content) and T$_{2m}$, and negative linear relationship between monthly averaged atmospheric pressure and T$_{2m}$. We also found negative relationship between atmospheric pressure and normalized SFs (Figure S8a). Considering air temperature always displays negative relationship with atmospheric pressure as warmer air temperature coincides with lighter air mas and lower atmospheric in summer, and colder air temperature coincides with heavier air mass and higher atmospheric pressure in winter. Hence, the temperature can be used to represent co-influence of both temperature and atmospheric pressure, and we only focus on the influence of temperature on CH$_4$ emissions and will add more supporting data in following studies." on lines 549-562.

Line 508: Please clarify that this is only the temperature component of the EF and does assume no changes in technology or other meteorological variables.

As answered above and to make clarification, we added "We should note the precipitation, soil water content and atmospheric pressure can also have obvious influence on CH$_4$ emissions, and considering the fact that we have not conducted field measurement in landfills and landfills are usually covered by metal or plastic in China to avoid the spread of odor smell, hence reanalysis data cannot represent real soil water contents in these site scale landfills. Precipitation and atmospheric pressure showed obvious linear relationship with temperature as displayed in Figure S8. They displayed positive linear relationship between precipitation (affect water content) and T$_{2m}$, and negative linear relationship between monthly averaged atmospheric pressure and T$_{2m}$. We also found negative relationship between atmospheric

pressure and normalized SFs (Figure S8a). Considering air temperature always displays negative relationship with atmospheric pressure as warmer air temperature coincides with lighter air mas and lower atmospheric in summer, and colder air temperature coincides with heavier air mass and higher atmospheric pressure in winter. Hence, the temperature can be used to represent co-influence of both temperature and atmospheric pressure, and we only focus on the influence of temperature on $CH_4$ emissions and will add more supporting data in following studies." on lines 549-562.

And also added "We should note that new technology and other meteorological variables can also influence waste treatment $CH_4$ emissions. The main reason to only use temperature in this study is that we only constrained the emissions at monthly scale in one year, and derived twelve datasets of *posteriori* $CH_4$ emissions. Besides, temperature is considered as the main factor in controlling monthly and annual variations of waste treatment $CH_4$ emissions, and can be used to represent co-influence of other meteorological parameters as atmospheric pressure. We will use multiple years' $CH_4$ concentration to quantify the influence of new technology and other meteorological variables on waste treatment $CH_4$ emissions in our following study, and we suggest other tracers (i.e. ethane, $^{14}CH_4$) are also important to separate $CH_4$ emissions from biological and fossil $CH_4$ emissions." on lines 677-686.

Line 529: Agreed that this is beyond the scope, but it seems prudent to mention that changes in management and technology can have a strong influence emissions in the future.
Done as suggested, please see the reply to comment above.

Line 570: What was the predicted emission change due to changes in activity data and management in the cited studies? How does you reported temperature sensitivity compare?
For the mentioned three cited studies (USEPA 2013; Cai et al., 2018; Spokas et al., 2021), USEPA (2013) and Cai et al. (2018) only predicted emission change due to changes in activity data and management technology. And the $CH_4$ emissions for year of 2030 by Cai et al. (2018) was 23.5% lower than USEPA (2013) estimation, which was caused by the consideration of new policies (NP) and low-carbon (LC) policy scenarios. And Spokas et al. (2021) modeled the $CH_4$ emission changes with increasing air temperature, where CH4 emissions did not show obvious changes even with temperature increased by ~5°C at the end of year 2100.

We added more explanation for clarification as "For the mentioned three cited studies, USEPA (2013) and Cai et al. (2018) only predicted emission change due to changes in activity data and management technology. And the $CH_4$ emissions for year of 2030 by Cai et al. (2018) was 23.5% lower than USEPA (2013) estimation, which was caused by the consideration of new policies and low-carbon policy scenarios. And Spokas et al. (2021) modeled the $CH_4$ emission changes with increasing air temperature, where $CH_4$ emissions did not show obvious changes even with temperature increased by ~5°C at the end of year 2100."

Line 606: Large parts of the conclusion sections are actually a summary.
Done as suggested, we revised this conclusion sections and deleted some sentences.

---

## Referee Report (RR1)

Review of revised manuscript by Cheng Hu, Junqing Zhang, Bing Qi, Rongguang Du, Xiaofei Xu, Haoyu Xiong, Huili Liu, Xinyue Ai, Yiyi Peng, and Wei Xiao

New title: Global warming will largely increase waste treatment $CH_4$ emissions in Chinese megacities: insight from the first city scale $CH_4$ concentration observation network in Hangzhou city, China

The authors have made many positive improvements to this manuscript, especially adding clarification and references. However, I continue to have significant questions.

1. Decomposition of organic waste by methanogens mostly takes at depth within the waste pile and temperatures can be significantly above those at the surface (https://www.atsdr.cdc.gov/hac/landfill/html/ch2.html). If the major types of cover are plastic and metal, both impervious to gas flow through the waste pile, then temperatures could get very high. However, other studies have found that there is a correlation between emissions and ambient temperature (https://rc.library.uta.edu/uta-ir/handle/10106/11641?show=full; CLEEN model) and between emissions and soil temperature (https://journals.sagepub.com/doi/abs/10.1177/0734242X9701500104?journalCode=wmra; Börjesson G, Svensson BH. Seasonal and Diurnal Methane Emissions From a Landfill and Their Regulation By Methane Oxidation. Waste Management & Research. 1997;15(1):33-54. doi:10.1177/0734242X9701500104). Thus, the relationship between emissions fluxes and temperature is very complicated and probably not simply related to ambient temperature. You might want to discuss this more, since temperature is so central to your conclusions.

2. Are there landfill gas collection systems at the facilities in and around Hangzhou? These are a management technique, but might they affect your thesis and conclusions?

3. I have concerns about your model's ability to simulate the nighttime PBLH (planetary boundary layer height), since this is very important to your use of 24-hour data. The plot that you show for your previous study in Nanjing only compares daytime measurements – there are no data from nighttime. If the model underestimates the nighttime PBLH, then it will underestimate nighttime emissions and, therefore, full diurnal emissions. Thus, the model would indicate artificially low scaling factors for the waste treatment sector. What do the results suggest if only daytime modeling and data are used?

4. You need to have a native English speaker proofread the manuscript. There are many places where it can be difficult to understand the meaning. I have made many suggestions in the attached Word document, but you need to make sure meaning wasn't changed mistakenly by these suggestions.

5. Should figure S8 be deleted from the Supplementary Information?

[revised manuscript text omitted]

Commented [A5]: Add a panel for pressure. It would be easier to see the comparison if you show either the differences between the observed and simulated meteorological parameters or scatter plots of simulated versus observed values..

Commented [A6]: Explain what these sites are - NOAA, other?

environmental                    science              and              data              center (http://www.resdc.cn/DOI),2017.DOI:10.12078/2017121101).

**2.2 WRF-STILT model setup**

The WRF-STILT (WRF: Weather Research and Forecasting, version 4.2.2, and STILT: Stochastic

Time-Inverted Lagrangian Transport) model will be used to simulate hourly footprint and $CH_4$

enhancement, see more details in Hu et al. (2019; 2021). Domain setups are displayed in Figure 1a, with the outer nested domain (Domian-1, 27 km×27 km grid resolution) covers covering eastern and central China, and the inner domain (Domain-2, 9 km×9 km grid resolution) covers covering the YRD area. The physical schemes used in the WRF model are the same as in our previous studies for the YRD domain (Hu et al., 2019; 2021). The simulated $CH_4$ concentration is the sum of background and enhancement, where the enhancement is calculated by multiplying all $CH_4$ flux with hourly footprint that represents the sensitivity of the concentration changes to its regional sources/sinks with spatial resolution of 0.1°×0.1°. To better quantify $CH_4$ components at each site,

$CH_4$ enhancements from different regions and sources are also tracked and separately simulated.

Besides, we should note the $CH_4$ background is important in simulating $CH_4$ concentrations and atmospheric inversion. We will choose $CH_4$ background from the five background sites based on monthly footprint as discussed in Section 3.1.

The most recent inventory of Emission Database for Global Atmospheric Research (EDGAR v6.0), which has 20 categories, and WetCHARTs ensemble mean were used as the *a priori* anthropogenic and natural $CH_4$ emissions. We should note there are many $CH_4$ inventories for some developed regions and countries (i.e. France, U.S.A., Germany) with high spatial resolutions. the The reasons to choose EDGAR as *a priori* anthropogenic emissions are: (1) for all available $CH_4$ inventories that covered China, the spatial resolution of EDGAR (0.1°×0.1°) is the highest, and it provides the most up-to date results; (2) most of previous studies that constrain emissions by atmospheric inversion studies also chosed EDGAR, and our results can be directly compared with previous studies; (3) the preliminary simulation of $CH_4$ concentrations showed generally good performance with observations,

Commented [A7]: Use "is" or "was", not "will be."

Commented [A8]: Section 2.1?

indicating its spatial distributions in Hangzhou city has relatively small bias even with a potentially large bias for magnitude, which will be constrained by our atmospheric inversion method.

The main sources of CH4 emissions in Hangzhou city include SWD_LDF (solid waste landfills),

WWT (waste water handling), SWD_INC (solid waste incineration), PRO (all processes related to fuel exploitation from coal, oil, and natural gas), RCO (energy for buildings, mainly containing nature gas escape escaping from household use) and AGS (agricultural soils). We found emissions from SWD_LDF, WWT

> **Commented [A9]:** Does this include extraction, refining, transportation, and burning for electricity generation of these fuels? Also includes fugitive emissions for natural gas? State this in the text, not just in the response to the comments. I assume that this is from fugitive emissions.

[revised manuscript text omitted]

besidesMoreover, waste treatment seems to emitted $CH_4$ by as area sources instead of point sources as from waste treatment super plants. Although a few previous studies found limitations of EDGAR

inventory to capture $CH_4$ emission patterns in some urban areas (Pak et al., 2021), here considering the fact that locations of landfills, which is the largest anthropogenic $CH_4$ emitter in Hangzhou city, are very close to the core urban area and in high consistence consistency with EDGAR, hence we believe the spatial patterns of EDGAR in study region can to be reliable. We should note the Chinese government constructed waste separation station in each city with density of one station for per

150~200 households (around 450~800 people), which can emit lots of methane caused by daily biomass waste as area sources (Tian et al., 2022). These above analyses also imply Hangzhou site can observe higher emissions from both waste treatment and total anthropogenic emissions, which will be discussed and quantified later.

**3.3 Simulation of $CH_4$ concentrations and its components for three sites**

Comparisons between observed and simulated daily $CH_4$ concentration averages are displayed in

Commented [A11]: Show where the landfills are located. You probably need to show an enlarged version of Hangzhou.

Commented [A12]: Are these in addition to the landfills? How long does material remain at these facilities before being transferred to landfills? Generally, not much methane is generated if the material only stays for a few days.

Figure 4a-c and hourly concentrations in Figure S4 for three sites. First, the hourly simulations in

Figure S4 showed high  consistency when only comparing the temporal patterns with observations, indicating good performance of model transport simulations as confirmed in Figure

S5 for evaluating meteorological fields. But the relative variations display obvious differences among the three sites for daily averages in Figure 4a-c. The mean bias (MB), root mean squared error (RMSE), and correlation coefficient (R) between daily observations and *a priori* simulations were 64.1 ppb, 129.2 ppb and 0.44, respectively, for Hangzhou site; and were -6.0 ppb, 57.1 ppb,

0.50 for Linan site, 36.2 ppb, 55.6 ppb, 0.54 for Damingshan site. As for the Hangzhou site, simulated $CH_4$ concentrations show obvious overestimation from October to April, and the overestimation was also found at Damingshan site. We found the simulations at the Linan site shows overall good agreement with observation, but still with slight overestimation from January to April and underestimation from May to September. Considering the source area contributions for the three sites are different, these difference among three sites indicated the bias in $CH_4$ emission largely varied from Hangzhou city to larger regional scale.

To further quantify detailed contributions from different regions and categories to each tower site,

$CH_4$ enhancements from different categories and source areas were also simulated separately for the three sites. As displayed in Figure 4d-e, the simulated *a priori* total enhancements at Hangzhou site,

Linan site, and Damingshan site were 244.3 ppb, 100.8, and 69.0 ppb, respectively. We also found contributions by waste treatment dominated the total enhancements but with obvious differences among the three sites, which varied from the highest 64.2% at Hangzhou site to the lowest

41.4% at Damingshan site. We further calculated anthropogenic contributions from Hangzhou city (excluding wetlands because of coarser spatial resolution for Hangzhou city) and other provinces, which were 158.4 ppb at Hangzhou site, 30.7 ppb at Linan site, and 10.1 ppb at Damingshan site, respectively. And they accounted for 69.3%, 34.0%, and 16.9% of total anthropogenic enhancements at corresponding sites. These results indicate the $CH_4$ observations at Hangzhou site, which is located at the core urban region, was more influenced by local emissions (mainly for waste treatment  which will be discussed later) and contain much higher enhancements than the other two sites. The relative contributions from different regions also imply that the observations at Linan

**Commented [A13]:** Especially for the Hangzhou and Daminshan site, the nighttime simulations frequently are higher than the observations, assuming that the high values are the nighttime values. This suggests that the simulation of nighttime boundary layer height is not so good, despite the relatively good meteorological comparisons in Figure S5. Scatter plots would show the relationships better - the deviations of the simulation-observation pairs from the 1:1 line. Indeed, the correlation coefficients are not great.

[revised manuscript text omitted]

**Commented [A18]:** Since you are concluding that temperature is controlling the SFs and EFs, please show the relationship between temperature and SFs, like you do for pressure and SFs.

**Commented [A19]:** Add a panel for atmospheric pressure in Figure S5, to illustrate this.

The spatial distribution of $T_{2m}$ trends for  all of China are displayed in Figure S10, which show heterogeneous distributions across China for four global warming scenarios.

Because east China has high population density , with the majority of the national population (Figure S1), and  is responsible for the largest domestic garbage induced $CH_4$ emissions (Figure S2), these combined factors indicate considerable $CH_4$

emissions changes from waste treatment in such a temperature-sensitivity area. Considering that the temperature sensitivity of waste treatment $CH_4$

EFs are caused by microbial process at  regional scales, it can represent general conditions of different cities or landfills. And if we assume the derived temperature sensitivity (increase by 44%

with temperature increase  of 10ºC on average) is applicable for  China as a whole, especially for east

China, the relative changes of waste treatment $CH_4$ EFs can be calculated by multiplying this value by air temperature trends.  The spatial distributions of global warming induced EF

changes at the end of this century are displayed Figure 9. For RCP2.6 scenario, EFs for waste treatment will slightly increase by 4.0-6.5% in  northern China and increase by 3.0-4.0%

in south east China. The RCP6.0 also displayed heterogeneous changes in east China, with

EFs in  north east China  increasing by 10.5-13.0% and in south east China increasing by 9.0-10.5%.

Relative changes in RCP4.5 and RCP8.5 are more homogeneous for east China, which indicates EFs will significantly increase by 5.0-7.5% and 17.5-19.5%, respectively. The largest changes will occur in west China for RCP8.5, with EFs increasing by >20.0%, but this area has low population density and $CH_4$ emissions, and  therefore these  effects of global warming can be ignored (Figure S10?). Finally, we should note these derived relative changes are only caused by global warming, and the influence of activity data, management technology and other factors is not considered and out of the scope of this study.

**4 Discussions and implications**

Many previous studies have compared total $CH_4$ emissions and its components for different inventories and bottom-up methods, which illustrated large uncertainty and bias at city scale and

**Commented [A20]:** Most of the microbial activity is occurring at depth in the decomposing waste pile, where surface temperature variations have very limited, if any effect.

**Commented [A21]:** What is "it"?

[revised manuscript text omitted]

Commented [A24]: Please enlarge the font of the text so it will be legible when the paper is printed.

[Figure]

Figure 5. Seasonal averaged diurnal variations for Hangzhou site in (a) winter, (b) spring, (c)
summer, (d) autumn, and Linan site in (e) winter, (f) spring, (g) summer, (h) autumn, and
Damingshan site in (i) winter, (j) spring, (k) summer, (l) autumn; Note because of two months of
data gap in Autumn for Damingshan site, the green line is for all September-November simulations,
red line only represent simulation of corresponding period for available observation data, and bold
lines represents data between 12:00 and 18:00.

[Figure]

Figure 6. Comparisons of hourly CH₄ concentrations at Hangzhou site between observations and
simulations by using (a) *a priori* and (b) *posteriori* emissions, (c) scatter plots of daily CH₄ averages
by using *a priori* and *posteriori* emissions.

[Figure]

Figure 7. (a) Monthly anthropogenic (excluding agricultural soil) CH$_4$ emissions for *a priori* and
*posteriori* emissions for Hangzhou city, (b) relationship between the monthly *posteriori* CH$_4$
emissions and temperature  for the three cases (differing uncertainties) discussed in section 2.3 of
the text.

[Figure]

Figure 8. (a) Annual air temperature from year 2021 to 2100 for four different global warming
scenarios for Hangzhou city, (b) the projected relative change of waste treatment $CH_4$ emissions (or
EFs) for Hangzhou city, note the shading indicates extent of three cases.

[Figure]

Figure 9. Global warming induced relative changes of waste treatment CH₄ EFs by year of 2100 for
(a) RCP2.6, (b) RCP4.5, (c) RCP6.0, and (d) RCP8.5 scenarios. Note the red boundary is Zhejiang
province.

Table 1. The *posteriori* SFs for different categories in three cases, where wetland: natural and 1135 agricultural wetland, Waste: waste treatment, PRO: fuel exploitation, RCO: energy for building, 1136 Others: the rest anthropogenic emissions.

| Month | Case 1 | | | ase 2 | | | | | Case 3 | | |
|---|---|---|---|---|---|---|---|---|---|---|---|
| | Wetland | Waste | Others | Wetland | Waste | PRO | RCO | Others | Wetland | Waste | Others |
| 1 | 1.00 | 0.29 | 0.83 | 1.00 | 0.34 | 0.90 | 0.80 | 0.93 | 1.00 | 0.40 | 0.72 |
| 2 | 1.00 | 0.20 | 0.89 | 1.00 | 0.26 | 0.97 | 0.83 | 0.93 | 1.00 | 0.30 | 0.77 |
| 3 | 1.03 | 0.39 | 1.04 | 1.02 | 0.46 | 1.07 | 0.80 | 0.97 | 1.02 | 0.46 | 0.95 |
| 4 | 1.10 | 0.46 | 0.96 | 1.08 | 0.48 | 1.01 | 0.95 | 0.93 | 1.08 | 0.49 | 0.91 |
| 5 | 1.12 | 0.62 | 0.99 | 1.10 | 0.64 | 1.06 | 0.97 | 0.92 | 1.11 | 0.65 | 0.95 |
| 6 | 1.22 | 0.59 | 1.09 | 1.18 | 0.64 | 1.05 | 0.97 | 1.03 | 1.18 | 0.64 | 1.05 |
| 7 | 1.10 | 0.88 | 0.96 | 1.09 | 0.88 | 1.00 | 1.00 | 0.94 | 1.09 | 0.89 | 0.94 |
| 8 | 1.05 | 0.62 | 0.95 | 1.01 | 0.66 | 0.99 | 0.97 | 0.95 | 1.01 | 0.67 | 0.91 |
| 9 | 1.04 | 0.71 | 1.01 | 1.02 | 0.73 | 0.96 | 0.98 | 1.04 | 1.02 | 0.74 | 0.98 |
| 10 | 1.06 | 0.60 | 0.94 | 1.06 | 0.61 | 0.92 | 0.96 | 1.00 | 1.06 | 0.62 | 0.90 |
| 11 | 1.01 | 0.27 | 0.86 | 1.00 | 0.32 | 0.91 | 0.85 | 0.93 | 1.00 | 0.37 | 0.75 |
| 12 | 1.00 | 0.31 | 0.70 | 1.00 | 0.33 | 0.75 | 0.79 | 0.91 | 1.00 | 0.43 | 0.58 |

Commented [A25]: Please list the cases again, so the reader doesn't have to search through the text.

Make sure you state in the title or caption that the values in this table are for the Hangzhou city site.

---

## Editor Decision (ED1)

5 March 2023

Dear authors

The reviewers point out the manuscript has much improved. However, there are still a number of issues that require further attention. For this please see the reviewers' reports and address all points in your response (and revised manuscript where appropriate). I want to stress a main point, namely the issue regarding the nighttime PBLH modeling. Please carefully address the question (point 3 : What do the results suggest if only daytime modeling and data are used ?) from the reviewer on this topic in your response and revised manuscript as I believe it is quite important. In addition, after applying the corrections to the english as provided by the reviewers, please have a native speaker go over your manuscript.

Ilse Aben

---

## Author Response (AR2)

**Reviewer 1:**

The authors have adequately responded to all points raised by this reviewer. The manuscript has been significantly improved and the focus of the study is now clearer.

The topic of Megacity GHG emissions in emerging economies of great importance and this study exemplifies important work on methane emissions in China. I recommend to publish the paper after minor corrections by the authors.

Thanks so much for the overall positive comments, we have further revised this MS following all your suggestions as follows.

Specific/technical comments:

line 38-39: consider rephrasing for clarity.

Done as suggested, we revised this sentence as "and considering the high temperature sensitivity of $CH_4$ emission factors (EFs) for the biological processes-based sources such as waste treatment, large bias will be caused when estimating future $CH_4$ emissions under different global warming scenarios.".

line 47: Suggest to change 'But' to 'In contrast, '

Done as suggested.

line 81: suggest to change 'ignorable' to 'negligible'

Done as suggested.

line 100: suggest to change 'supposed' to 'expected'

Done as suggested.

line 102 and elsewhere: change 'east China' to 'East China'

Done as suggested.

line 113: change 'approach' to 'approaches'

Done as suggested.

line 126: same as line 113

Done as suggested.

line 129: change 'retrieval owns' to 'observations have'

Done as suggested.

line 131: change 'nearly' to 'recently'

Done as suggested.

line 141: change 'in' to 'on'

Done as suggested.

line 144: consider removing 'by' and 'and' for clarity

Done as suggested, we have revised this sentence as "There was only one recent study which focused on urban

waste treatment CH$_4$ emissions".

line 151: please clarify if you are referring to one specific bottom-up approach or to bottom-up approaches in general here.

We have revised it as "compared with "bottom-up" approaches, the "top-down" method can avoid using the factors that lead to large uncertainties of CH$_4$ emissions especially from waste treatment".

line 230: consider reformatting
Done as suggested.

line 250: change 'provide' to 'provides'
Done as suggested.

line 326: it remains unclear how this uncertainty was calculated
Here the uncertainty is calculated based on the assumption that monthly CH$_4$ concentration in September and October varied between August and November, we have revised this sentence as "where the uncertainty is calculated based on the assumption that monthly CH$_4$ concentration in September and October varies between August and November" on lines 332-333.

line 358: change 'retain' to 'remain'
Done as suggested.

line 362: change 'shown' to 'shows'
Done as suggested.

line 408: change 'present' to 'represent'
Done as suggested.

line 419: please correct grammar
Done as suggested, we have revised it as "And total CH$_4$ emissions will be overestimated when using daytime emissions to represent all-day averages".

line 437: remove 'by'
Done as suggested.

line 471: remove 'by'
Done as suggested.

line 510 to 514: consider rephrasing for clarity
Done as suggested, we have revised this sentence as "After including the constraints from the observed concentrations, the *posteriori* emissions for waste treatment show obvious seasonality with highest emission in July ($7.66 \pm 0.09 \times 10^3$ t) and lowest emission in February ($2.20 \pm 0.87 \times 10^3$ t). And emissions from other anthropogenic categories show much smaller seasonality (highest emission in January of $4.18 \pm 0.69 \times 10^3$ t and lowest emission in August of $2.88 \pm 0.15 \times 10^3$ t) than *a priori* emissions." on lines 537-542.

line 610: remove '-wide'

Done as suggested.

line 615: consider changing 'above' to 'reported'

Done as suggested.

line 637: remove 'mentioned'

Done as suggested.

line 643+: please correct grammar

Done as suggested, we have revised this sentence as "To our best knowledge, there are no inventories that considered the temperature-induced changes on both seasonal variations and annual trends of $CH_4$ emissions."

line 653: change 'not' to 'no'

Done as suggested.

line 687: Still a lot of repetition and summarizing in this section. Suggest to change title to 'Summary and Conclusions'

Done as suggested.

Overall, the manuscript switches between past tense and present. Please consider harmonizing the writing.
There is also an overabundant use of 'displays/displayed'

Done as suggested, we have used present tense for these words.

**Reviewer 2:**

Review of revised manuscript by Cheng Hu, Junqing Zhang, Bing Qi, Rongguang Du, Xiaofei Xu, Haoyu Xiong, Huili Liu, Xinyue Ai, Yiyi Peng, and Wei Xiao New title: Global warming will largely increase waste treatment CH4 emissions in Chinese megacities: insight from the first city scale CH4 concentration observation network in Hangzhou city, China The authors have made many positive improvements to this manuscript, especially adding clarification and references. However, I continue to have significant questions.

Thanks so much for these detailed suggestions. All points have been addressed below (review query in Italic; author response in blue). Changes to the text in the manuscript have been marked in blue.

1. Decomposition of organic waste by methanogens mostly takes at depth within the waste pile and temperatures can be significantly above those at the surface (https://www.atsdr.cdc.gov/hac/landfill/html/ch2.html). If the major types of cover are plastic and metal, both impervious to gas flow through the waste pile, then temperatures could get very high. However, other studies have found that there is a correlation between emissions and ambient temperature (https://rc.library.uta.edu/uta-ir/handle/10106/11641?show=full; CLEEN model) and between emissions and soil temperature (https://journals.sagepub.com/doi/abs/10.1177/0734242X9701500104?journalCode=wmra; Börjesson G, Svensson BH. Seasonal and Diurnal Methane Emissions From a Landfill and Their Regulation By Methane Oxidation. Waste Management & Research. 1997;15(1):33-54. doi:10.1177/0734242X9701500104). Thus, the relationship between emissions fluxes and temperature is very complicated and probably not simply related to ambient temperature. You might want to discuss this more, since temperature is so central to your conclusions.

Thanks so much for pointing it out, we agree that temperature within landfills should be much more related to methanogens activities and $CH_4$ emissions than $T_{2m}$. However, considering (1) we do not have direct temperature observations under landfills, (2) only $T_{2m}$ is provided for future climate RCP scenarios, and (3) air temperate can act as good indicator of the general variations for landfill temperature, hence the relationship between waste $CH_4$ emissions and $T_{2m}$ is constructed and used to predict how does $CH_4$ emission changes in different RCP scenarios.

To make clarification, we also added more descriptions as "Note decomposition of organic waste by methanogens mostly takes at depth within the landfills and temperature can be higher than at the surface, hence the temperature within landfills should be much more related to methanogens activities and $CH_4$ emissions than $T_{2m}$. However, considering (1) we do not have direct temperature observations under landfills, (2) $T_{2m}$ can be used as indicator of methanogens activities, and (3) $T_{2m}$ is commonly used meteorological data that can be provided for future RCP scenarios, hence the relationship between waste $CH_4$ emissions and $T_{2m}$ is constructed and used to predict how will $CH_4$ EFs change in different climate scenarios." on lines 572-579.

2. Are there landfill gas collection systems at the facilities in and around Hangzhou? These are a management technique, but might they affect your thesis and conclusions?

There are two main methods to deal with waste in Hangzhou, with the first one burning waste and second one by landfills. And for the second one, we should note the Chinese government constructed waste separation station in each city with density of one station for per 150~200 households (around 450~800 people), usually these waste separation stations are full with waste because domestic garbage can be generated every day, they do not have gas collection systems and can emit large quantity of $CH_4$ emissions caused by daily biomass waste as area sources (Tian et al., 2022). Besides, there is only one landfill that has gas collection systems, the reported gas collection efficiency was less than 80%, which also indicates large quantity of $CH_4$ emissions will be directly emitted into the atmosphere and the emissions will be influenced by climate change.

To make clarification, we have added more descriptions on lines 377-384 as "We should note the Chinese government constructed waste separation stations in each city with density of one station for per 150~200 households (around 450~800 people), usually these waste separation stations are full with waste because domestic garbage can be generated every day, they do not have gas collection systems and can emit large quantity of $CH_4$ emissions caused by daily biomass waste as area sources (Tian et al., 2022). Besides, there is only one landfill that has gas collection systems, the reported gas collection efficiency was less than 80%, which also indicates large quantity of $CH_4$ emissions will be directly emitted into the atmosphere and the emissions will be influenced by climate change.".

We also revised the sentence on lines 114-117 as "For these "bottom-up" approaches, the high uncertainties were directly attributed to omission of many small point sources and discrepancies of observed site-specific EFs, which varied largely by climate and management technology such as the efficiency of gas collection systems".

3.I have concerns about your model's ability to simulate the nighttime PBLH (planetary boundary layer height), since this is very important to your use of 24-hour data. The plot that you show for your previous study in Nanjing only compares daytime measurements ´ there are no data from nighttime. If the model underestimates the nighttime PBLH, then it will underestimate nighttime emissions and, therefore, full diurnal emissions. Thus, the model would indicate artificially low scaling factors for the waste treatment sector. What do the results suggest if only daytime modeling and data are used?

[Figure]

Figure R1. Comparisons between simulated and observed hourly PBLH in (a) December 2020, (b) May 2021, (c) July 2021, and (d) September 2021.

Done as suggested. Firstly, we have contacted with the environmental protection department of Hangzhou city, which provides us with four months (one month in each season) of hourly PBLH observations in study period, and the comparisons between simulated and observed PBLH in each month represent general good performance of WRF model. As displayed in Figure R1, it shows overall good performance for both daytime and nighttime PBLH variations, and indicates our WRF-STILT model can well simulate atmospheric transport processes as found in our previous studies. Secondly, we also only used daytime (11:00-16:00, local time) $CH_4$ observations and simulations to derive *posteriori* scaling factors and analyzed the temperature sensitivity. Besides, we further analyzed the

temperature sensitivity of $CH_4$ EFs by only using daytime $CH_4$ observations and simulations in Figure S10, it still shows strong linear relationship between normalized SFs and $T_{2m}$, with the slopes of 0.046 and 0.060. These results are in high consistency with using all-day observations of 0.038 and 0.050, indicating similar results of using 24 hours observations and only using daytime observations, and less influence of simulated nighttime PBLH bias on corresponding temperature sensitivity.

To make clarification, we have added "Note PBLH simulations are important in evaluating model performance, we only have four months of PBLH observations (one month in each season), these hourly PBLH observations were used to evaluate the general performance of WRF model. As displayed in Figure S6, it shows overall good performance for both daytime and nighttime PBLH variations." on lines 461-465. All related tables and figures have been added in supplementary file.

Besides, as suggested by reviewer, we also only used daytime $CH_4$ observations and simulations to derive *posteriori* scaling factors and analyzed the temperature sensitivity. Considering results from Case 2 varied between Case 1 and Case 3, here we only displayed the results from Case 1 and Case 3 in Table R1 and Figure R2. It shows strong linear relationship between temperature, with the slopes of 0.046 and 0.060, these results are similar with using all 24 hours data of 0.038 and 0.050.

We added "Although the evaluations of hourly PBLH simulations have illustrated good performance in both daytime and nighttime (Figure S6), we also conducted inversions by only using daytime observations to constrain $CH_4$ emissions. Considering results from Case 2 varied between Case 1 and Case 3, here we only display the results from Case 1 and Case 3 (Table S2), it shows similar seasonal variations as using all all-day observations. We notice the values are larger than later, which is reasonable because $CH_4$ emissions in daytime should be larger than all-day and nighttime emissions. In general, *posteriori* SFs by using all-day concentration observations will be used to represent total $CH_4$ emissions from monthly to annual scales." on lines 504-512.

Table R1. *Posteriori* Scaling factors by only using daytime (11:00-16:00, local time) observations and simulations.

| Month | Case 1 | | | Case 3 | | |
|---|---|---|---|---|---|---|
| | Wetland | Waste | Others | Wetland | Waste | Others |
| 1 | 1.00 | 0.64 | 0.71 | 1.00 | 0.54 | 0.78 |
| 2 | 1.00 | 0.44 | 0.82 | 1.00 | 0.33 | 0.93 |
| 3 | 1.01 | 0.56 | 0.98 | 1.02 | 0.48 | 1.06 |
| 4 | 1.12 | 0.73 | 1.13 | 1.13 | 0.70 | 1.16 |
| 5 | 1.16 | 1.23 | 1.01 | 1.15 | 1.26 | 0.97 |
| 6 | 1.08 | 0.98 | 1.18 | 1.08 | 0.98 | 1.18 |
| 7 | 1.04 | 1.36 | 1.41 | 1.01 | 1.42 | 1.36 |
| 8 | 1.02 | 0.95 | 0.99 | 1.03 | 0.94 | 1.00 |
| 9 | 1.02 | 0.75 | 0.97 | 1.04 | 0.71 | 1.01 |
| 10 | 1.08 | 0.68 | 1.02 | 1.08 | 0.65 | 1.05 |
| 11 | 1.00 | 0.37 | 0.72 | 1.00 | 0.23 | 0.83 |
| 12 | 1.00 | 0.59 | 0.66 | 1.00 | 0.42 | 0.74 |

[Figure]

Figure R2. Relationship between the monthly *posteriori* $CH_4$ emissions and temperature in case 1 and 3, where the emissions are constrained by only using daytime(11:00-16:00, local time) observations.

We also added "We also analyzed the temperature sensitivity by only using daytime $CH_4$ observations and simulations in Figure S10, it still shows strong linear relationship between normalized SFs and $T_{2m}$, with the slopes of 0.046 and 0.060. These results are in high consistency with using all-day observations of 0.038 and 0.050, indicating similar results of using 24 hours observations and only using daytime observations, and less influence of simulated nighttime PBLH bias on corresponding temperature sensitivity." on lines 582-587.

4. You need to have a native English speaker proofread the manuscript. There are many places where it can be difficult to understand the meaning. I have made many suggestions in the attached Word document, but you need to make sure meaning wasn't changed mistakenly by these suggestions.

Thanks so much and we really appreciate your help especially for providing these detailed suggestions on corresponding English grammars, we have revised these typos and also asked one native English speaker to proofread it.

5. Should figure S8 be deleted from the Supplementary Information?

Here we added both daily *posteriori* and *priori* $CH_4$ concentrations to be compared with observation, which aims to illustrate the improvement of $CH_4$ emissions.
